# Pseudo-Nonlinear Data Augmentation: A Constrained Energy Minimization Viewpoint

**Pingbang Hu**[*]
University of Illinois Urbana-Champaign
pbb@illinois.edu

**Mahito Sugiyama**
National Institute of Informatics & SOKENDAI
mahito@nii.co.jp

## Abstract

We propose a simple yet novel data augmentation method for general data modalities based on energy-based modeling and principles from information geometry. Unlike most existing learning-based data augmentation methods, which rely on learning latent representations with generative models, our proposed framework enables an intuitive construction of a geometrically aware latent space that represents the structure of the data itself, supporting efficient and explicit encoding and decoding procedures. We then present and discuss how to design latent spaces that will subsequently control the augmentation with the proposed algorithm. Empirical results demonstrate that our data augmentation method achieves competitive performance in downstream tasks compared to other baselines, while offering fine-grained controllability that is lacking in the existing literature.

## 1 Introduction

*Data augmentation* has advanced considerably in recent years, driven largely by the increasing use of generative models (Kingma & Welling, 2014; Chadebec et al., 2022; Antoniou, 2017; Trabucco et al., 2024) to meet the demand for larger and more diverse datasets (Feng et al., 2021; Wong et al., 2016). Beyond traditional domains such as images, these methods have been extended to a wide range of modalities. Despite their promise, however, generative-model-based augmentation faces several fundamental challenges. First, data augmentation is most valuable when training data is scarce, yet in such cases, we typically lack a pre-trained foundational model for the target domain. This creates a paradox: before we can augment the data, we must first train a generative model—reintroducing the very problem of limited data. Second, even when suitable foundational models are available, the computationally intensive nature of deep generative methods poses practical obstacles. Since effective augmentation often requires generating data of the same order as the original dataset, the cost of large-scale generation can quickly become prohibitive. Third, augmenting data with generative models raises concerns about their interpretability and controllability (Guidotti et al., 2018). Consequently, even when these models perform well, the lack of understanding of the underlying transformations of the augmented data makes it difficult to control the generated outputs, which poses a significant risk in the case of high-stakes scenarios (Rudin, 2019).

In this work, we propose a new data augmentation framework that addresses the challenges outlined above by providing a **learning-free**, **efficient**, and **controllable** algorithm applicable across diverse data modalities. Our approach builds on the well-established theory of *energy-based models* (Xie et al., 2016), together with recent advances in *log-linear models on partially ordered sets (posets)* (Sugiyama et al., 2016; 2017) and *information geometry* (Amari, 2016; Amari & Nagaoka, 2000; Ay et al., 2017). Conceptually, our framework resembles an autoencoder (Kingma & Welling, 2014). We begin by parametrizing data as discrete probability distributions on a *curved* statistical manifold $\mathcal{S}$. The data is then *encoded* into a chosen "latent space" $\mathcal{B} \subseteq \mathcal{S}$ via *forward projection*. Within this latent space, simple augmentation procedures informed by the encoded data are applied. Finally, the resulting "augmented representation" is *backward projected* to the local data space $\mathcal{D} \subseteq \mathcal{S}$, yielding new augmented data. As our proposed algorithm exploits the duality of the projection in the statistical manifold $\mathcal{S}$, where it is linear in the manifold's intrinsic coordinates yet non-linear in the ambient space, we hence term it *pseudo-non-linear* data augmentation.

---

[*]Work done in part while the author was visiting National Institute of Informatics.

This design offers three key advantages. First, it is learning-free: the sub-manifold structure is constructed explicitly, allowing direct control over the properties of the augmented data without the need to train a generative model. Second, it is computationally efficient: both forward and backward projections can be formulated as convex programs and solved with efficient first-order methods such as gradient descent. Third, it provides controllability: leveraging prior knowledge about relationships among features, one can adjust the choice of $\mathcal{S}$ and the sub-manifold of projection to tailor the statistical properties of the augmented data. Our contributions are summarized as follows:

- We propose a novel framework for modeling structured data (e.g., tensors) within a statistical manifold using energy-based models. This framework captures the intrinsic geometry of data and enables the design of geometry-aware algorithms.
- We develop the *pseudo-nonlinear* data augmentation algorithm under this framework. The method is **learning-free**, **efficient**, and **controllable**, and applies broadly across different data modalities.
- We empirically validate the effectiveness of our approach, showing that it achieves competitive or superior performance compared to both generative-model-based baselines (e.g., autoencoders) and classical augmentation methods across multiple datasets and modalities.

## 2 RELATED WORK

### 2.1 LEARNING-BASED DATA AUGMENTATION

Data augmentation has proven to be effective in enhancing deep learning training by increasing dataset size, improving model robustness (Rebuffi et al., 2021), and introducing implicit regularization (Hernández-García & König, 2018). These techniques have been applied across various modalities, including text (Shorten et al., 2021; Feng et al., 2021; Li et al., 2022a) and images (Shorten & Khoshgoftaar, 2019; Mumuni & Mumuni, 2022; Wang et al., 2017). Due to the generality and the popularity, much of the recent progress in data augmentation for general modalities has been driven by advancements in generative models, such as autoencoders (Kingma & Welling, 2014; Chadebec et al., 2022), generative adversarial networks (Antoniou, 2017), and diffusion models (Trabucco et al., 2024). Despite the progress, to date, there is no fully satisfactory solution for the two challenges mentioned for generative-model-based data augmentation, i.e., efficiency and controllability. For example, the design of controllable GANs (Li et al., 2022b; She et al., 2021) and efficient flow-based models (Geng et al., 2025) remains an active area of research, and exploration in this direction is largely limited to specific domains such as images.

### 2.2 LEARNING-FREE DATA AUGMENTATION

Learning-free data augmentation methods (Maharana et al., 2022) that do not rely on generative models are particularly appealing because they construct an explicit, low-dimensional "latent space" in which data can be represented and augmented with fine-grained and intuitive control and interpretability. These latent spaces are typically derived from classical *dimension reduction* techniques such as Principal Component Analysis (PCA)(Wold et al., 1987) and Singular Value Decomposition (SVD)(Stewart, 1993). In general, these methods identify an optimal *linear* subspace and then perform augmentations that respect this (Euclidean) geometry of the ambient subspace. Beyond their simplicity and transparency, linear methods also provide useful insights; for instance, PCA reveals principal directions that capture the dominant modes of variation in the data.

A major limitation of linear dimension reduction for augmentation, however, is the difficulty of the *inverse* problem: reconstructing high-dimensional data from low-dimensional representations without a learned decoder is often non-trivial. Some works attempt to circumvent this issue by leveraging linear dimension reduction only indirectly for augmentation (Abayomi-Alli et al., 2020; Sirakov et al., 2024). Other approaches, such as mixup (Zhang et al., 2018), bypass dimension reduction entirely and operate directly in the original data ambient space. While popular in practice due to simplicity, these methods are typically heuristic, application-specific, and harder to generalize.

*Non-linear* generalizations of dimension reduction is often referred to as *manifold learning* (Meilă & Zhang, 2024), which provides an alternative route. Methods such as t-SNE (Hinton & Roweis, 2002; Van der Maaten & Hinton, 2008), Isomap (Tenenbaum et al., 2000), and UMAP (McInnes et al., 2018) aim to exploit the manifold hypothesis, which posits that high-dimensional data lie

near a lower-dimensional manifold embedded in the ambient space. Their goal is to uncover this manifold and produce a smooth embedding that captures the intrinsic geometry of the data.

In principle, manifold learning could avoid the inverse problem by recovering a low-dimensional manifold with minimal information loss, making it an attractive candidate for data augmentation. In practice, however, this goal is rarely achieved without incorporating learning mechanisms (Duque et al., 2020; Coifman & Lafon, 2006; Williams & Seeger, 2000; Vladymyrov & Carreira-Perpinán, 2013; Han et al., 2022). A workaround is to exploit the latent space learned by a model already trained on a downstream task, as in manifold mixup (Verma et al., 2019). Yet these approaches again sacrifice interpretability and controllability, compared to fully learning-free augmentation methods.

## 3 PRELIMINARY

### 3.1 DUALLY-FLATNESS IN INFORMATION GEOMETRY

Information geometry studies the structure of *statistical manifolds* $\mathcal{S}$ within the space of probability distributions. In this paper, we are primarily concerned with the space of an exponential family $\{p_\theta(x) \mid \theta \in \mathbb{R}^D\}$, where each $p_\theta$ denotes a probability density function parameterized by $\theta$. We focus on the key concept in this field, *dually-flatness*, in this preliminary, while directing readers to Section A and Amari (2016) for more comprehensive details.

The starting point is the observation that the *log-partition function* $\psi(\theta)$ (also known as the *cumulant generating function* in statistics and *free energy* in physics) of an exponential family with density $p_\theta$ is convex in the *natural parameter* $\theta \in \mathbb{R}^D$. This convexity induces a natural coordinate system, $\theta$, on $\mathcal{S}$, defining both the Riemannian metric $g = \nabla^2 \psi(\theta)$ and the Bregman divergence (Bregman, 1967) $D_\psi(p_\theta, p_{\theta'})$. With these structures, the manifold $(\mathcal{S}, g)$ is flat, meaning that any curve $\theta(t) = at + b$ (where $a, b \in \mathbb{R}^D$ are constants) is a geodesic and lies entirely within $\mathcal{S}$. This flatness is known as *e-flatness*, and the geodesics are referred to as *e-geodesics* or *primal-geodesics*.

The dual structure arises from the *Legendre transform* (Legendre, 1787), which generates the dual function $\psi^*(\eta)$, where $\eta \in \mathbb{R}^D$ is the *expectation parameter*. This dual function is also convex, giving rise to the expectation coordinate system $\eta$, the dual Riemannian metric $g^*$, and also the dual Bregman divergence $D_{\psi^*}$ which is the well-known Kullback-Leibler divergence $D_{\mathrm{KL}}$ (Eq.(3)). The corresponding flatness is termed *m-flatness*, with *m-geodesics* or *dual-geodesics* as its geodesics.

**Remark 3.1.** *An e-flat (m-flat) sub-manifold can be defined by forcing linear constraints on the $\theta$ coordinates ($\eta$ coordinates) (Amari, 2016, Chapter 2).*

*Dually-flatness* emerges from the interplay between these two structures. Specifically, for any point (distribution) $p$ in $\mathcal{S}$, there is a unique point $p^*$ on an e-flat sub-manifold $\mathcal{B} \subseteq \mathcal{S}$ that minimizes the dual Bregman divergence $D_{\psi^*}(p, q) = D_{\mathrm{KL}}(p, q)$ (Amari, 2016, Theorem 1.5). This process, known as the *m-projection*, can be efficiently solved via convex optimization (Section A.2). The dual holds when switching $e$ and $m$. Projection is a central tool in information geometry with profound implications for understanding the geometry of $\mathcal{S}$, which we will use later.

### 3.2 STATISTICAL MANIFOLD ON POSETS

A set $\Omega$ is a *partially ordered set (poset)* if it is equipped with a *partial order* "$\leq$", a relation satisfying the following for all $x, y, z \in \Omega$: 1.) $x \leq x$ (reflexivity); 2.) $x \leq y$ and $y \leq x$ implies $x = y$ (antisymmetry); 3.) $x \leq y$ and $y \leq z$ implies $x \leq z$ (transitivity). We focus on finite posets $\Omega$ with a bottom element $\perp$ such that $\perp \leq x$ for all $x \in \Omega$ to prevent some technical difficulties.

Given such a poset $\Omega$, consider a discrete random variable $X$ with finite support $\Omega$ with its probability mass function $p \colon \Omega \to \mathbb{R}_{\geq 0}$, $p(x) = \Pr(X = x)$ for $x \in \Omega$. For a discrete probability distribution $p$ over a poset $\Omega$, the *log-linear model on posets* recursively defines $\theta \colon \Omega \to \mathbb{R}$ as $\log p(x) =: \sum_{y \leq x} \theta(y)$ for all $x \in \Omega$ (Sugiyama et al., 2017). This model belongs to the exponential family, with $\theta$ corresponding to the natural parameters, except for $\theta(\perp)$, which coincides with the partition function. Thus, all discrete probability distributions over $\Omega$ form a $(|\Omega| - 1)$-dimensional dually-flat statistical manifold $\mathcal{S} := \{p \colon \Omega \to \mathbb{R}_{\geq 0} \mid \sum_{x \in \Omega} p(x) = 1\}$, with the dual coordinate systems $(\theta, \eta)$ depend on the poset structure.

**Remark 3.2.** *One can think of $\theta(x)$ for each element $x \in \Omega$ as specifying the* energy *(i.e., $p(x)$) for $x$, and the relation between $\theta(x)$'s on different elements $x$'s is specified by the poset structure.*

## 4 PSUEDO-NONLINEAR DATA AUGMENTATION

We first present our proposed framework in Section 4.1 and the projection algorithms in Section 4.2, then we combine and apply them to data augmentation in Section 4.3. Finally, we discuss two important features of the proposed method regarding **controllability** and **efficiency** in Section 4.4. Throughout this section, we will use *positive tensors* as our running example (Theorem 4.1).

### 4.1 LOG-LINEAR MODEL ON POSETS FRAMEWORK

Motivated by Theorem 3.2, by associating each element $x \in \Omega$ with a feature dimension, we can specify the geometry among features—i.e., design the poset structure—using prior knowledge or natural structure present in the data. The resulting models define an energy-aware, curved statistical space that faithfully reflects the prescribed relationships among features.

More specifically, given a dataset $\{z_i\}_{i=1}^n$, we first embed the data into a statistical manifold $\mathcal{S}$ by leveraging the log-linear model on posets, which provides a geometric structure induced by the energy-based modeling. The process works in three steps: 1.) models each $z_i$ as a *real-valued poset*; 2.) embeds the real-valued poset into the statistical manifold $\mathcal{S}$ by viewing it as a probability distribution; 3.) computes the corresponding two coordinate representations using the log-linear model on posets. See Figure 1 for an illustration. We now explain each step in detail below.

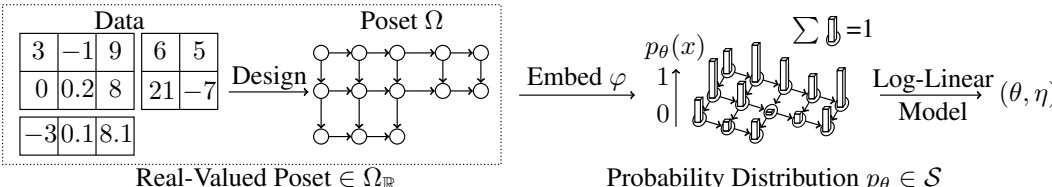

Figure 1: Given structured data, we first design a poset $\Omega$ that reflects the structure or the relationship between features. The resulting real-valued poset is then embedded into the statistical manifold $\mathcal{S}$ as a discrete probability distribution $p_\theta(x)$ via an embedding $\varphi$. Finally, the log-linear model on posets provides the dually-flat coordinates $(\theta, \eta)$ for $p_\theta$, which can be computed efficiently (Section 3.2).

**Real-Valued Poset.** In the typical machine learning pipeline, inputs are often constrained to be vectors or matrices, which fail to accommodate more complex data. In contrast, posets are flexible enough to capture data with structures, including vectors, matrices, or tensors. In general, any data structure that admits a natural partial order can be modeled by a poset: for instance, a $D$-dimensional vector $z \in \mathbb{R}^D$ (i.e., 1$^{\text{st}}$-order tensor) admits a structure that can be modeled by the poset $\Omega := [D]$ with the partial order being the natural order between positive integers. Similarly, other common data structures, such as matrices or tensors, can be treated in the same way. On the other hand, as noted at the beginning of the section, a custom poset structure $\Omega$ can also be specified, either with or without the natural structure, to reflect prior knowledge of the relation between features.

We can then define the *real-valued poset*, which is a mapping from the poset $\Omega$ to the set of real numbers $\mathbb{R}$ such that each entry (element) of the data structure (poset) $x \in \Omega$ is associated with a feature in $\mathbb{R}$. We denote the set of real-valued posets as $\Omega_\mathbb{R}$. In the $D$-dimensional vector example, $\Omega = [D]$, each element $x \in \Omega$ corresponds to one of the $D$ dimensions. Associating a real number to each dimension then corresponds to an element in $\Omega_\mathbb{R}$.

**Embedding.** To embed the data $\{z_i \in \Omega_\mathbb{R}\}_{i=1}^n$, which are now modeled as real-valued posets, to the statistical manifold $\mathcal{S}$ which concerns with discrete probability distributions, we want an embedding $\varphi \colon \Omega_\mathbb{R} \to \mathcal{S}$ and its inverse $\varphi^{-1} \colon \mathcal{S} \to \Omega_\mathbb{R}$, such that $\sum_{x \in \Omega}(\varphi(z_i))_x = 1$ for all $z_i$ with $\dim(\mathcal{S}) = D - 1$.[1] In other words, $\varphi(z_i)$ gives a probability mass function of a discrete random

---

[1]One can also consider the manifold of positive measures of dimension $D$ and avoid the potential scaling issues. For simplicity, we omit this trivial extension in the presentation.

variable over the poset, where $(\varphi(z_i))_x$ is the probability of sampling $x \in \Omega$ when sampled from $\varphi(z_i)$, representing the *energy* of the feature $x$. From the perspective of energy-based modeling, as noted in Theorem 3.2, $\varphi$ can be naturally induced, e.g., for tabular frequency data, or customized to reflect the desired energy relationship between features. We note that in the latter case, a joint design of $\varphi$ with the poset structure $\Omega$ is often more expressive to describe a desirable energy structure.

**Dually-Flat Coordinates.** Finally, from the log-linear model on posets introduced in Section 3.2, for each $z_i' := \varphi(z_i) \in \mathcal{S}$, we associate the dually-flat coordinates $\theta(z_i') \in \mathbb{R}^{D-1}$ and $\eta(z_i') \in \mathbb{R}^{D-1}$. Such coordinate systems are defined with respect to the underlying poset structure $\Omega$, providing a representation that captures the prescribed geometric structure among features of the data. We next illustrate the framework with one canonical example, positive tensors, in Theorem 4.1:

**Example 4.1** (Positive tensor). *A $d^{th}$-order tensor $T \in \mathbb{R}^{I_1 \times \cdots \times I_d} =: \mathbb{R}^D$ is a multidimensional array with real entries for every index vector $v = (i_1, \ldots, i_d) \in [I_1] \times \cdots \times [I_d] =: \Omega$ where for each $k$, $[I_k] := \{1, 2, \ldots, I_k\}$ for a positive integer $I_k$. Tensors with entries all being positive are called positive tensors, denoted as $P \in \mathbb{R}_{\geq 0}^{I_1 \times \cdots \times I_d}$.*

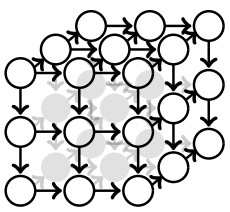

*A natural partial order "$\leq$" between two index vectors $v = (i_1, \ldots, i_d)$, $w = (j_1, \ldots, j_d)$ for tensors is that $v \leq w$ if and only if $i_k \leq j_k$ for all $k \in [d]$. Finally, for positive tensors, a simple embedding $\varphi \colon \mathbb{R}_{\geq 0}^{I_1 \times \cdots \times I_d} \to \mathcal{S}$ where $P' := \varphi(P) \colon \Omega \to \mathbb{R}_{\geq 0}$ such that $P_v' := P_v / \sum_{w \in \Omega} P_w$ for all $v \in \Omega$ can be defined with a natural empirical inverse (see Theorem 4.3).*

Figure 2: Natural poset structure of $3^{\text{rd}}$-order tensors in $\mathbb{R}^{3 \times 3 \times 3}$.

Theorem 4.1 applies to many common data modalities. For example, a color image can be represented as a $3^{\text{rd}}$-order tensor, where the first two dimensions correspond to height and width, and the third dimension encodes the color channels. Time-series data likewise admit a tensorial poset structure, with the temporal dimension inducing a natural ordering among features at discrete time steps. Despite this flexibility, our framework comes with one notable limitation:

**Remark 4.2** (Invariance). *Because the framework relies on specifying a partial order over the index set, it is not naturally equipped to model invariances under index permutations. This can introduce unwanted bias, for instance, in settings such as graphical data. Nonetheless, a key advantage of being learning-free and fully transparent is that the source of such bias is explicit, allowing one to identify and, when desired, mitigate it by appropriately modifying the modeling choices.*

To conclude, we emphasize that the notion of *energy* in our framework acts as a modality-specific potential function that quantifies the stability or plausibility of a data configuration under the chosen poset embedding. Once the poset is specified, both the energy function and the induced manifold structure follow directly, yielding a clean and computationally convenient representation that supports subsequent computations and algorithm design, which we will discuss next.

## 4.2 FORWARD AND BACKWARD PROJECTION

We now demonstrate how to incorporate the log-linear on poset framework with projection theory to conduct data augmentation. Our algorithm mimics the architecture of autoencoders, focusing on two building blocks: the *encoder* $\mathsf{Enc}(\cdot)$ and the *decoder* $\mathsf{Dec}(\cdot)$. First, for the encoding step, we formally explain how projection theory can be applied to perform dimension reduction and obtain compact representations within our framework. Next, for the decoding step, we introduce our proposed algorithm, termed *backward projection*, which serves as the *inverse* of dimension reduction. Figure 3 provides an intuitive geometric illustration for our proposed algorithms.

**Forward Projection.** The embedding from $\Omega_{\mathbb{R}}$ to $\mathcal{S}$ introduced in Section 4.1 maintains the dimensionality. To achieve dimension reduction,

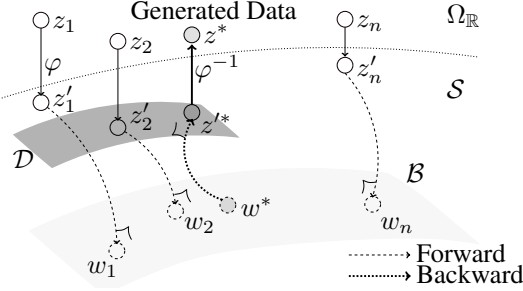

Figure 3: Illustration of forward and backward projection. Here, $w_i$: latent representation of the original data $z_i$, obtained from forward projection to $\mathcal{B}$; $w^*$: augmented latent representation; $w^* \mapsto z'^*$: backward projection to $\mathcal{D}$, obtained from the original data of the nearest neighbor(s) of $w^*$ in the latent space.

we leverage the projection theory: by projecting $z_i' = \varphi(z_i)$ onto a low-dimensional flat sub-manifold called *base sub-manifold* $\mathcal{B} \subseteq \mathcal{S}$ with $\dim(\mathcal{B}) \ll \dim(\mathcal{S})$, we obtain the desired *encoding* $\mathsf{Enc} := \mathrm{Proj}_{\mathcal{B}} \circ \varphi \colon \Omega_{\mathbb{R}} \to \mathcal{B}$ that maps the data to a low-dimensional latent representation. Note that the encoding $\mathsf{Enc}(\cdot)$ is smooth and well-defined as the projection is unique when $\mathcal{B}$ is flat and minimizing either the primal or the dual Bregman divergence, depending on either $\mathcal{B}$ is $e$- or $m$-flat.

**Backward Projection.** One of the technical burdens is that the encoding $\mathsf{Enc}(\cdot)$ is not invertible, hence a perfect decoding $\mathsf{Dec}(\cdot)$ is mathematically impossible, even when $\mathsf{Enc}(\cdot)$ only involves a simple linear projection in Euclidean space. Here, we propose a simple, geometrically intuitive, and data-centric solution that aims to find the *inverse* of the projection with theoretical guarantees.

The high-level intuition is simple: we assume that similar data will result in similar projections. Hence, given a point in the low-dimensional latent representation space, we try to "project it back" to approximate the original dataset by exploiting the fact that we have access to the inverse of the dataset's projection, which is the dataset itself. Specifically, we can artificially create a sub-manifold $\mathcal{D}$ around a subset of the dataset that captures the local geometric structure of the dataset around the latent representation, and *backward* project onto it.

Formally, assuming that we have access to the embedded dataset $\{z_i' = \varphi(z_i)\}_{i=1}^n$ and their projected result $\{w_i = \mathrm{Proj}_{\mathcal{B}}(z_i')\}_{i=1}^n$ for some flat base sub-manifold $\mathcal{B}$. To find the inverse of some given point $w^* \in \mathcal{B}$ assuming it comes from the projection on $\mathcal{B}$, we first find $w^*$'s $k$-nearest neighbors among $w_i$'s, obtaining a size $k$ index set $N \subseteq [n]$ with $|N| = k$. Then we create a flat sub-manifold $\mathcal{D}$ called *local data sub-manifold* based on the pre-images $z_i'$'s of these $w_i$'s, and project $w^*$ on $\mathcal{D}$ to obtain the *inverse* $z'^* = \mathrm{Proj}_{\mathcal{B}}^{-1}(w^*) := \mathrm{Proj}_{\mathcal{D}}(w^*)$. The construction of $\mathcal{D}$ can be arbitrary, in particular, one can easily control the degree of freedom of the resulting $z'^*$: for instance, from Theorem 3.1, given the nearest neighbor $z_{i^*}'$, one can define an $e$-flat $\mathcal{D} := \{\theta \in \mathbb{R}^{\dim(\mathcal{S})} \mid (\theta)_x = (\theta(z_{i^*}'))_x \text{ for some } x \in \Omega\}$ by fixing some indexes of $\theta$ to be the corresponding $\theta$-coordinate values of $z_{i^*}'$.

Algorithm 1 in Section B summarizes this procedure, which we termed *backward projection*. With access to $\mathrm{Proj}_{\mathcal{B}}^{-1}(\cdot)$, decoding is simply $\mathsf{Dec} := \varphi^{-1} \circ \mathrm{Proj}_{\mathcal{B}}^{-1} \colon \mathcal{B} \to \Omega_{\mathbb{R}}$. Algorithm 1 is a geometrically intuitive, data-centric algorithm with desirable theoretical guarantees such as divergence minimizing when projecting on the constructed local data sub-manifold $\mathcal{D}$.

### 4.3 Psuedo-Nonlinear data augmentation

With all the building blocks in place, we can now formally describe the proposed data augmentation algorithm, which consists of: 1.) encoding, 2.) augmenting, and 3.) decoding.

**Encoding.** As described in Section 4.2, the encoding $\mathsf{Enc} := \mathrm{Proj}_{\mathcal{B}} \circ \varphi$ is simply a combination of the embedding followed by a projection. Notation-wise, we write $w_i := \mathsf{Enc}(z_i)$.

**Augmenting.** To generate an augmented data $z^*$, we first generate a new representation $w^*$ in the latent space, which in our case, is a pre-specified flat base sub-manifold $\mathcal{B}$. This $w^*$ can be generated in various ways, such as controlled perturbations of the original representations or a linear mixture of two arbitrary original representations.

**Decoding.** As described in Section 4.2, the decoding $\mathsf{Dec} := \varphi^{-1} \circ \mathrm{Proj}_{\mathcal{B}}^{-1}$ is simply a combination of backward projection (Algorithm 1) with the inverse of the embedding. Notation-wise, we write $z^* := \mathsf{Dec}(w^*) = \varphi^{-1}(z'^*)$ where $z'^* := \mathrm{Proj}_{\mathcal{B}}^{-1}(w^*) := \mathrm{Proj}_{\mathcal{D}}(w^*)$.

The proposed method integrates the *nonlinear* forward and backward projections as encoding and decoding, which we summarize the above in Algorithm 2 with an illustration given by Figure 3.

**Remark 4.3** (Positive tensor). *With Algorithm 2, the empirical inverse $\varphi^{-1}$ for the positive tensors in Theorem 4.1 can be defined as the* inverse of the average of the scaling among nearest neighbors.

### 4.4 Sub-manifolds for positive tensors

It is evidence that the proposed method is **learning-free**. In this section, we describe how to construct flat sub-manifolds (for $\mathcal{B}$ and $\mathcal{D}$) to **control** the augmentation process, and how these sub-manifolds naturally admit **efficient projection algorithms**. For clarity and concreteness, we focus our discussion on the case of positive tensors, while noting that the principles and arguments extend to broader settings for other data modalities, poset structures, and the design of the energy potential.

**Designing Sub-Manifolds.** We start by discussing an intrinsic trade-off of choosing the dimension of $\mathcal{B} \subseteq \mathcal{S}$ we aim to forward project on. Clearly, more information about the data is preserved after forward projection onto $\mathcal{B}$ as $\dim(\mathcal{B})$ increases. Hence, the quality of the backward projection $\mathrm{Proj}_{\mathcal{B}}^{-1}(\cdot)$ (Algorithm 1) increases along with $\dim(\mathcal{B})$. However, in the extreme case when $\dim(\mathcal{B}) \approx \dim(\mathcal{S})$, Algorithm 2 becomes less effective as the augmenting step now suffers from the curse of dimensionality. Previous studies on such a trade-off of choosing $\dim(\mathcal{B})$ (Sugiyama et al., 2018; Ghalamkari et al., 2024) reveal how one should construct flat base sub-manifolds. In particular, the *many-body tensor approximation* (Ghalamkari et al., 2024; Derun & Sugiyama, 2025) aims to capture a *hierarchy* of mode interactions with different $\dim(\mathcal{B})$ for positive tensors within the log-linear model on posets. Specifically, the $\ell$-body approximation considers projection on the following sub-manifold:

$$\mathcal{M}_\ell := \{\theta \in \mathbb{R}^{\dim(\mathcal{S})} \mid \theta_x = 0 \text{ for all } \textbf{non } \ell\text{-body parameters } x \in \Omega\}, \tag{1}$$

where the $\ell$-body parameter corresponds to $\ell$ non-one indices, acting as a generalization of one-body and two-body parameters (Ghalamkari et al., 2024). Intuitively speaking, an $\ell$-body parameter captures the interaction among $\ell$ different modes. Hence, when $\mathcal{B} = \mathcal{M}_\ell$, all interactions between modes of orders higher than $\ell$ are neglected. This approach provides a principled way of designing the latent space with a clear understanding of what each dimension signifies.

On the other hand, a dual-like trade-off exists for the local data sub-manifold $\mathcal{D}$. Recall that the goal of backward projection is to project back to the "local data" space $\mathcal{D}$ given by a set $N$ of $k$ nearest neighbors of a generated latent representation. When $\dim(\mathcal{D})$ increases, backward projecting onto $\mathcal{D}$ has a higher degree of freedom, which is desirable for data augmentation. However, in the extreme case when $\dim(\mathcal{D}) \approx \dim(\mathcal{S})$, the backward projection becomes unconstrained and potentially generates gibberish results. Hence, a natural construction of $\mathcal{D}$ is to consider the "dual" notion of $\mathcal{M}_\ell$, where we now allow all **non** $\ell$-body parameters to vary while fixing every $\ell$-body parameter to be the average of the $\theta$ values among $N$:

$$\mathcal{M}_\ell^*(N) := \left\{\theta \in \mathbb{R}^{\dim(\mathcal{S})} \mid \theta_x = \frac{1}{k} \sum_{i^* \in N} \left(\theta(z'_{i^*})\right)_x \text{for all } \ell\text{-body parameters } x \in \Omega\right\}. \tag{2}$$

These two constructions offer a practical design choice for Algorithm 2 while providing the desired properties. For instance, By choosing an appropriate $\ell$, both $\mathcal{M}_\ell$ and $\mathcal{M}_\ell^*$ can capture specific information with desired degree of freedom.

**Efficient Projection.** For sub-manifolds $\mathcal{M}_\ell$ designed for many-body approximation with $B$ many non-fixed indexes (i.e., $\ell$-body parameters), the projection can be efficiently computed via formulating the projection as a convex program over $B$ variables that can be solved via gradient descent in polynomial time (in terms of $B$). We note that the projection in the statistical manifold offers other desirable theoretical guarantees, such as minimizing the KL-divergence (corresponding to energy minimization) and uniqueness of the projection. Moreover, in the case of many-body approximation, it admits an efficient algorithmic implementation, since the gradient of the corresponding convex program has a closed-form, which makes the optimization extremely efficient. We refer the reader to Section A.2 for a detailed discussion.

## 5 EXPERIMENTS

The structure of this section is as follows. Firstly, Section 5.1 demonstrates the energy aspect of the proposed framework, validating our motivation. Then, Section 5.2 illustrates the proposed pseudo-nonlinear data augmentation on real-world visual datasets. Finally, we apply the pseudo-nonlinear data augmentation on downstream tasks in Section 5.3.

### 5.1 LOG-LINEAR MODEL WITH ENERGY INTERACTION

**Setup.** We demonstrate our proposed log-linear model on the posets framework with simple synthetic data under controlled feature interactions. The goal of this section is to illustrate both the energy-modeling perspective of our proposed framework, as well as how designing base sub-manifolds $\mathcal{B}$ allows us to capture higher-order mode interactions among data.

In particular, we model both the 2-dimensional intensity data and 1-dimensional time-series data as a $2^{nd}$-order tensor as described in Theorem 4.1, where we apply reshaping to time-series data to capture higher-order mode dependencies across the temporal dimension.

**Experiment.** Firstly, we consider **separable** and **non-separable** 2-dimension *intensity* data $I$. Separable data (Figure 4(a)) corresponds to data *without* feature interaction, where we generate $I(i, j) \approx f(i) \times g(j)$ for mode $i$ and $j$. On the other hand, non-separable data (Figure 4(b)) corresponds to data *with* intricate feature interaction, where $I(i, j)$ has higher-order interaction between modes $i$ and $j$ such that $I(i, j)$ is not factorizable.[2] From Figure 4, we remark two interesting properties that are central

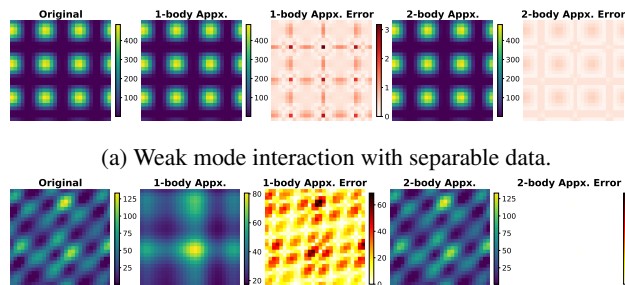

(a) Weak mode interaction with separable data.

(b) Strong mode interaction with non-separable data.

Figure 4: Mode interaction and approximation.

to our framework: 1.) we can *control* the order of mode interaction preservation intuitively with appropriate sub-manifold design; 2.) even when the base sub-manifold does not have enough capacity to capture the mode interactions (e.g., 1-body approximation in Figure 4(b)), it nevertheless captures the essence of the data within its capacity in the energy-minimizing sense.

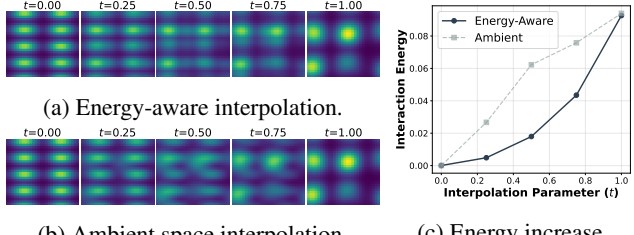

(a) Energy-aware interpolation.

(b) Ambient space interpolation.

(c) Energy increase.

Figure 5: Interpolation energy in different geometries.

Following a similar setup, we conduct a case study in Figure 5 to illustrate the advantages of our method's induced geometry compared to the vanilla ambient space geometry. In particular, we **interpolate** between separable and non-separable data within both the *base sub-manifold* (Figure 5(a)) and *ambient space* (Figure 5(b)), and measure the *interaction energy*: the discrepancy between the 1-body approximation 2-body approximation of the interpolated data. The result is shown in Figure 5(c), where we see that energy-aware base sub-manifold consistently induces a lower interaction energy compared to the ambient geometry, indicating that our method, when operating on top of which, utilizes less energy compared to its ambient space geometry counterpart.

Finally, in Figure 6, we consider time series data with temporal dependencies emerging from higher-order interactions. A similar trend as Figure 4 emerges: the latent space geometry derived from a simple base sub-manifold results in a larger error at the place where higher-order mode interactions are presented (i.e., high frequency chirp pattern). In contrast, an appropriate base sub-manifold can capture such a mode interaction perfectly.

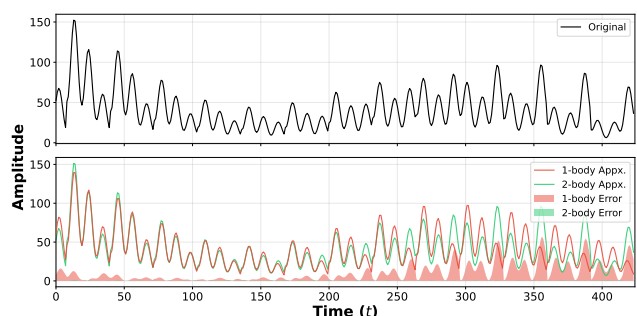

Figure 6: Approximation of temporal-dependent time-series.

## 5.2 CONTROLLABILITY WITH CHOICES OF SUB-MANIFOLDS

**Setup.** As discussed in Section 4.4, constructing the sub-manifold carefully provides the essential controllability. We demonstrate this with MNIST (LeCun, 1998) and CIFAR (Krizhevsky & Hinton, 2009), where we first apply the log-linear model on posets for positive tensors (Theorem 4.1) by

---

[2]Equivalently, separable data is *linear* in log space where $\log I(i, j) = \log f(i) + \log g(j)$, while non-separable data has non-linear mode interactions in log space.

normalizing features to be positive and reshaping the features as a tensor of suitable dimensions, then apply our proposed pseudo-nonlinear data augmentation afterwards. In particular: 1.) for MNIST, we let $\mathcal{B} = \mathcal{M}_1$ and $\mathcal{D} = \mathcal{M}_1^*$, which correspond to preserving *shape* information; 2.) for CIFAR, we first carefully *reshape* the colored images to higher-order tensors, and let $\mathcal{B} = \mathcal{M}_5$ and $\mathcal{D} = \mathcal{M}_4^*$, which correspond to preserving *fine-grained collective shape and color* information.

**Experiment.** Figures 7(a) and 7(d) show the results of the forward projection, while Figures 7(b) and 7(e) show the results of the backward projection of the latent representations sampled from the kernel density model $M$ fitted on the results ($\theta$-coordinates) of Figures 7(a) and 7(d), respectively.

For MNIST, we see that the augmentation results (Figure 7(b)) with backward projection successfully reconstruct the digit structures, indicating that the essential *shape* information is indeed preserved and separated in the latent space $\mathcal{B}$ to provide non-trivial neighbor information for constructing a sufficiently good $\mathcal{D}$ for backward projection. Note that the local data sub-manifold $\mathcal{D}$ has a dimension of 767, indicating a high degree of freedom for backward projection.

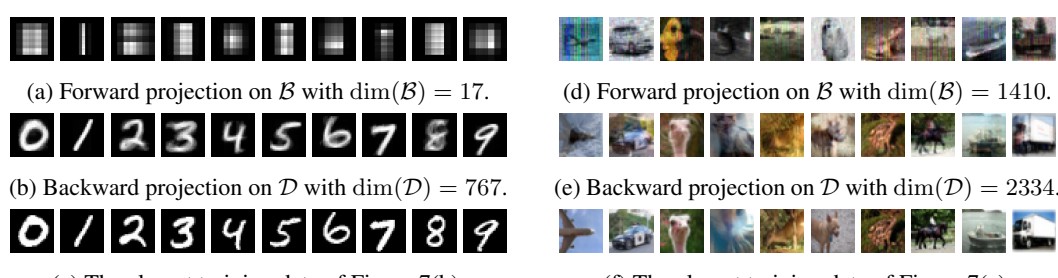

(a) Forward projection on $\mathcal{B}$ with $\dim(\mathcal{B}) = 17$.      (d) Forward projection on $\mathcal{B}$ with $\dim(\mathcal{B}) = 1410$.

(b) Backward projection on $\mathcal{D}$ with $\dim(\mathcal{D}) = 767$.      (e) Backward projection on $\mathcal{D}$ with $\dim(\mathcal{D}) = 2334$.

(c) The closest training data of Figure 7(b).      (f) The closest training data of Figure 7(e).

Figure 7: Results of MNIST (Left) and CIFAR-10 (Right) for Algorithm 2. The first row shows some representative latent representations from the dataset, while the second row shows the backward projection from an *augmented* latent representation.

More interesting results for CIFAR-10 are shown in Figures 7(d) to 7(f). By our proposed projection-based augmentation method, the *fine-grained shape and color* information is preserved. For instance, the third image, ostrich, successfully preserves the fine-grained shape and color relationship (e.g., colors for eyes and beak, and small pink flowers in the background), while the crude shape-to-color information is lost (e.g., colors for the background without shape details shift noticeably). The same trend can be observed consistently, validating the proposed method's efficacy.

In practice, by carefully reshaping the data into higher-order tensors such that some modes of the tensors contain the essential relationship between features that one wishes to control, with many-body approximation, it is possible to construct suitable sub-manifolds that preserve the chosen information, providing a controllable augmentation of the original data via simple projection operations.

## 5.3 CLASSIFICATION PERFORMANCE

**Setup.** We apply our proposed pseudo-nonlinear data augmentation algorithm on downstream classification tasks across different datasets with various modalities, including image (MNIST and CIFAR-10), audio (Speech Commands (Warden, 2018)), and tabular data (Connectionist Bench (Sejnowski & Gorman, 1988), Taiwanese Bankruptcy (Journal, 2020), and Wine Quality (Cortez et al., 2009)). For each dataset, we train a classifier on both the original training set and an augmented training set, where the augmented portion corresponds to 20% of the original training size and is generated from the training data. The classifiers used are ResNet-18 (He et al., 2016) for CIFAR-10, M5 (Dai et al., 2017) for SpeechCommands, and a simple MLP for the remaining datasets. All of the models are evaluated on 20 randomly bootstrapped test subsets, each containing 50% of the original test data. Further details of the training setup are provided in Section C.1.

**Experiment.** Several baselines are compared against our proposed method, including both learning-based and learning-free methods, including: 1.) pseudo-nonlinear (**PNL**, ours), 2.) autoencoder-based augmentation (**AE**) (Kingma & Welling, 2014; Chadebec et al., 2022),[3] 3.) mixup

---

[3]More recent learning-based baselines, e.g., diffusion-based augmentation (Trabucco et al., 2024), are often computationally infeasible at our scale and can only be applied in a "few-shot" setting.

Table 1: Test accuracy of classifiers trained on different datasets.

| Training Set | Dataset | | | | | |
|---|---|---|---|---|---|---|
| | MNIST | CIFAR-10 | Speech Commands | Connectionist Bench | Taiwanese Bankruptcy | Wine Quality |
| **OG** | $97.98 \pm 0.19\%$ | $88.57 \pm 0.57\%$ | $84.48 \pm 0.50\%$ | $88.10 \pm 8.58\%$ | $96.54 \pm 0.56\%$ | $55.00 \pm 1.69\%$ |
| **OG$^{\text{STD}}$** | $97.98 \pm 0.24\%$ | $89.89 \pm 0.44\%$ | $82.98 \pm 0.50\%$ | $85.24 \pm 7.66\%$ | $96.17 \pm 0.57\%$ | $57.85 \pm 1.81\%$ |
| **OG$^{\text{PNL}}$** | $97.91 \pm 0.21\%$ | $88.07 \pm 0.46\%$ | $84.35 \pm 0.37\%$ | $93.81 \pm 4.54\%$ | $96.53 \pm 0.47\%$ | $59.03 \pm 1.74\%$ |
| **OG$^{\text{AE}}$** | $97.97 \pm 0.25\%$ | $88.36 \pm 0.46\%$ | $83.13 \pm 0.32\%$ | $82.86 \pm 7.59\%$ | $95.92 \pm 0.62\%$ | $57.23 \pm 1.67\%$ |
| **OG$^{\text{MU}}$** | $96.45 \pm 0.23\%$ | $86.60 \pm 0.49\%$ | $81.85 \pm 0.61\%$ | $89.29 \pm 4.97\%$ | $96.55 \pm 0.68\%$ | $57.76 \pm 1.67\%$ |
| **OG$^{\text{MMU}}$** | $97.52 \pm 0.30\%$ | $88.02 \pm 0.39\%$ | $83.06 \pm 0.54\%$ | $91.19 \pm 5.06\%$ | $96.44 \pm 0.53\%$ | $58.70 \pm 1.74\%$ |

(**MU**) (Zhang et al., 2018) 4.) manifold mixup (**MMU**) (Verma et al., 2019), and 5.) standard augmentation (**STD**). For images, **STD** includes standard techniques such as random cropping, flipping, rotations, and affine transformations. For speech, **STD** combines random volume scaling, time stretching, MelSpectrogram conversion, frequency masking, and time masking (Park et al., 2019). For other data types, **STD** is implemented as Gaussian noise perturbations.

Results are summarized in Table 1, where we denote the original dataset as **OG**, and the augmented dataset using a method **AG** as **OG$^{\text{AG}}$**. In most cases, the classifier trained on the augmented data achieves a better prediction accuracy compared to the one trained only on **OG**. One exception is the dataset associated with image modality, where all other baselines perform worse than **OG** and **OG$^{\text{STD}}$**. A potential explanation is that the baseline augmentation algorithms for other baselines act more like regularizers, while **OG$^{\text{STD}}$** explicitly forces the classifier to learn the robust visual features associated with modality-specific transformation, promoting properties such as rotation, location, and color invariance. Nevertheless, in general, **PNL** consistently outperforms other baselines on all datasets, demonstrating a competitive performance across modalities.

Importantly, we highlight the stability, a crucial but often overlooked goal in data augmentation. This is best illustrated by the Connectionist Bench dataset, which contains only 208 data points and 60 features, posing a significant challenge for stable training and testing for generalization. From Table 1, the standard deviations in accuracy on this dataset are noticeably higher for **OG**, **OG$^{\text{STD}}$**, and **OG$^{\text{AE}}$**. In contrast, our method achieves a substantially lower standard deviation of $4.54\%$, indicating improved consistency across bootstrap testing runs. Other data augmentation baselines, such as **OG$^{\text{MU}}$** and **OG$^{\text{MMU}}$**, also achieve a substantial improvement. Still, our method achieves the lowest standard deviation among all, and this trend is consistently presented across all datasets.

### 5.4 Additional Experiments

We conduct a series of additional ablation studies and experiments in Sections C.2 to C.5. Specifically, Sections C.2 and C.3 assess the robustness of our proposed method and the impact of augmentation on downstream task performance. In contrast, Section C.4 provides a justification for the necessity of forward projection. Finally, Section C.5 explores the effect of different latent space design choices on augmentation outcomes, offering insights into how these design decisions can be leveraged to better control the augmentation process.

## 6 Conclusion

We introduced the *pseudo-nonlinear data augmentation* framework, which leverages information geometry and energy-based models to provide a **learning-free**, **efficient**, and **controllable** augmentation method. Our approach, grounded in the log-linear model on posets, endows data with a rich information-geometric structure induced from the designed energy potential, facilitating both geometric reasoning and principled algorithm design. A key component is the backward projection algorithm, which reverses dimension reduction in a geometrically intuitive and data-centric manner.

Through extensive experiments, we demonstrated the effectiveness of our method across diverse modalities and datasets. In particular, it enables scalable augmentation for general data types while offering controllability via the design of 1.) the base sub-manifold $\mathcal{B}$, 2.) the local data sub-manifold $\mathcal{D}$, and 3.) the poset structure $\Omega$. Empirically, our framework outperforms both learning-based and learning-free data augmentation baselines, even on common modalities such as images and audios.

## ACKNOWLEDGEMENTS

This work was supported by JSPS, KAKENHI Grant Number JP25H01112, JP25H01124, Japan and JST, CREST Grant Number JPMJCR22D3, Japan.

## ETHICS STATEMENT

We have carefully reviewed and adhered to the ICLR Code of Ethics. This work does not involve human subjects, personally identifiable information, or sensitive data. All datasets used in our experiments are publicly available and commonly used in the community. We have followed standard practices for dataset preprocessing and model evaluation to avoid introducing unfair bias. To the best of our knowledge, our methodology and findings do not pose potential harm, nor do they raise concerns regarding privacy, security, discrimination, or misuse.

## REPRODUCIBILITY STATEMENT

We are committed to ensuring the reproducibility of our results. The main paper provides a detailed description of the proposed method and the experimental setup. All hyperparameters, training procedures, and evaluation metrics are documented in Section C.1. To further facilitate reproducibility, we provide the source code and instructions for reproducing all experiments, which are publicly available at `https://github.com/sleepymalc/Pseudo-Nonlinear`.

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

# A PROJECTION THEORY IN INFORMATION GEOMETRY

We will assume some familiarity with the basic terminologies for manifold (Lee, 2012, Chapter 1, 4). In particular, in this section, we explain the main concepts of information geometry used in this study, including natural parameters, expectation parameters, model flatness, and convexity of optimization. In the following, we consider only discrete probability distributions for simplicity and refer to Amari (2016) for more general cases.

## A.1 $(\theta, \eta)$-COORDINATE AND GEODESICS

Consider $\mathcal{S}$ as the space of discrete probability distributions, which is a non-Euclidean space with the Fisher information matrix $G$ as the metric. This metric measures the distance between two points, i.e., discrete probability distributions, in $\mathcal{S}$. In Euclidean space, the shortest path between two points is a straight line, while in a non-Euclidean space, such a shortest path is called a *geodesic*. In the space $\mathcal{S}$, two kinds of geodesics can be introduced: $e$-geodesics and $m$-geodesics. For two points $p_1, p_2 \in \mathcal{S}$, $e$- and $m$-geodesics are defined as

$$\{r_t \mid \log r_t = (1-t)\log p_1 + t\log p_2 - \phi(t), 0 \le t \le 1\}, \quad \{r_t \mid r_t = (1-t)p_1 + tp_2, 0 \le t \le 1\},$$

respectively, where $\phi(t)$ is a normalization factor to keep $r_t$ to be a distribution.

We can parameterize distributions $p \in \mathcal{S}$ by parameters known as *natural parameters*. In Section 3.2, we have described the relationship between a distribution $p$ and a natural parameter vector $\theta \in \mathbb{R}^{D-1}$ for a discrete probability distribution over a sample space of $D$ elements in the log-linear model. The natural parameter $\theta$ serves as a coordinate system of $\mathcal{S}$, since any distribution in $\mathcal{S}$ is specified by determining $\theta$. Furthermore, we can also specify a distribution $p$ by its expectation parameter vector $\eta \in \mathbb{R}^{D-1}$, which corresponds to expected values of the distribution and an alternative coordinate system of $\mathcal{S}$. More explicitly, the definition of the expectation parameter $\eta$ is defined as $\eta_x = \sum_{y \ge x} p(y)$ for $x \in \Omega$, and $\eta_\perp = 1$, where $p(x)$ is the probability mass function of $p$ over the sample set $\Omega$, which is assumed to be a poset. The $\theta$-coordinates and $\eta$-coordinates are orthogonal with each other, which means that the Fisher information matrix $G$ has the following property, $G_{u,v} = \partial\eta_u/\partial\theta_v$, and $(G^{-1})_{u,v} = \partial\theta_u/\partial\eta_v$. $e$- and $m$-geodesics can also be described using these parameters as follows:

$$\{\theta_t \mid \theta_t = (1-t)\theta^{p_1} + t\theta^{p_2}, 0 \le t \le 1\}, \quad \{\eta_t \mid \eta_t = (1-t)\eta^{p_1} + t\eta^{p_2}, 0 \le t \le 1\},$$

where $\theta^p$ and $\eta^p$ are $\theta$- and $\eta$-coordinate of a distribution $p \in \mathcal{S}$.

## A.2 FLATNESS, PROJECTION, AND ITS OPTIMIZATION

A subspace is called $e$-*flat* when any $e$-geodesic connecting two points in a subspace is included in the subspace. The vertical descent of an $m$-geodesic from a point $p \in \mathcal{S}$ onto $e$-flat subspace $\mathcal{B}_e$ is called $m$-projection. Similarly, $e$-projection is obtained when we replace all $e$ with $m$ and $m$ with $e$. The flatness of subspaces guarantees the uniqueness of the projection destination. The projection destination $\bar{p}$ or $\tilde{p}$ obtained by $m$- or $e$-projection onto $\mathcal{B}_e$ or $\mathcal{B}_m$ minimizes the following KL divergence

$$\bar{p} = \arg\min_{q \in \mathcal{B}_e} D_{\text{KL}}(p, q), \quad \tilde{p} = \arg\min_{q \in \mathcal{B}_m} D_{\text{KL}}(q, p),$$

where the KL divergence from discrete distributions $p \in \mathcal{S}$ to $q \in \mathcal{S}$ is given as

$$D_{\text{KL}}(p, q) = \sum_{x \in \Omega} p(x) \log \frac{p(x)}{q(x)}, \tag{3}$$

where $p(x)$ and $q(x)$ are the probability mass functions of $p$ and $q$, respectively. A subspace with some of its natural parameters fixed at $0$ is $e$-flat (Amari, 2016, Chapter 2), which is obvious from the definition of $e$-flatness. More generally, any subspace $\mathcal{B}$ resulting from linear constraints on the natural parameter is $e$-flat. Similarly, any subspace $\mathcal{B}$ resulting from linear constraints on the expectation parameter is $m$-flat. When a space is $e$-flat and $m$-flat at the same time, we say that the space is *dually-flat*. The set of discrete probability distributions $\mathcal{S}$ is dually-flat.

Both $e$- and $m$-flatness guarantee that the cost functions to be optimized in Eq. (3) are convex. Therefore, $m$- and $e$-projection onto an $e$- or $m$-flat subspace can be implemented by a gradient

method using a second-order gradient. This second-order gradient method is known as the *natural gradient method* (Amari, 1998). The Fisher information matrix $G$ appears by second-order differentiation of the KL divergence. For instance, given $p$ and an $e$-flat subspace $\mathcal{B}_e$, the optimization problem $\overline{p} = \arg\min_{q \in \mathcal{B}_e} D_{\mathrm{KL}}(p, q)$ can be efficiently solved via gradient descent with second-order derivative by the update rule $\theta_{t+1} = \theta_t - G^{-1}(\eta_t - \eta^p)$, where $G \in \mathbb{R}^{D \times D}$ is the Hessian matrix, and $\partial D_{\mathrm{KL}}(P, Q)/\partial\theta = \eta - \eta^p$ is the derivative of the KL divergence. The updated natural parameters $\theta_{t+1}$ can then be used to construct $q_{t+1} \in \mathcal{B}_e$ that is closer to the destination $\overline{p}$ along with the $e$-geodesic from $q_t$ to $\overline{p}$. By repeating this process until convergence, we can always find the global optimal solution. A similar algorithm can be implemented for the other case, i.e., $\widetilde{p} = \arg\min_{q \in \mathcal{B}_m} D_{\mathrm{KL}}(q, p)$.

We make some remarks on the optimization of many-body approximation (Ghalamkari et al., 2024) that we omit in Section 4.4, which is a specific case of the above discussion.

**Example A.1** (Many-body approximation). *For $\mathcal{B}_e = \mathcal{M}_\ell$ defined in Eq. (1):*[4]

1. Convexity and uniqueness*: The solution of the many-body approximation is always unique, and the objective function of the many-body approximation is convex (Ghalamkari et al., 2024, Theorem 1). In particular, the many-body approximation is a maximum likelihood estimation that approximates a non-negative tensor, which is regarded as an empirical distribution, by an extended Boltzmann machine without hidden variables.*

2. Computational complexity*: The computational complexity of the many-body approximation for $\mathcal{B}_e = \mathcal{M}_\ell$ with $B$ many non-fixed indexes (i.e., $\ell$-body parameters) is $O(T|B|^3)$, where $T$ is the number of iterations of the optimization. This is because the overall complexity is dominated by the update of $\theta$, which includes matrix inversion of $G$, and the complexity of computing the inverse of an $n \times n$ matrix is $O(n^3)$. As an alternative, one can appeal to first-order methods such as gradient descent, which will then reduce the complexity to $O(T|B|)$, where $|B|$ corresponds to computing the gradient.*
   *Note that this complexity can be reduced if one reshapes tensors so that the size of each mode becomes small. We explore this idea further in Section C.5.*

## B  OMITTED DETAILS FROM SECTION 4

We provide the pseudocode for the proposed algorithm in Algorithms 1 and 2.

---

**Algorithm 1:** Backward projection

---

**Data:** A representation $w^* \in \mathcal{B}$, $\varphi$-embedded dataset $\{z_i'\}_{i=1}^n$ with projection $\{w_i\}_{i=1}^n$ on $\mathcal{B}$, $k \in \mathbb{N}$
**Result:** Backward projected data $z'^*$

$N \leftarrow$ `Nearest-Neighbor(`$k$, $w^*$, $\{w_i\}_{i=1}^n$`)`
$\mathcal{D} \leftarrow$ `Construct-Sub-Manifold(`$\{z_i'\}_{i \in N}$`)`
$z'^* \leftarrow$ `Proj(`$w^*$, $\mathcal{D}$`)`
**return** $z'^*$

---

---

[4]One can consider $\mathcal{B}_m$ with Eq. (1) being defined w.r.t. the $\eta$-coordinate system, and similar remarks hold.

---

**Algorithm 2:** Pseudo-non-linear data augmentation

---

**Data:** A dataset $\{z_i\}_{i=1}^n$, embedding $\varphi\colon \Omega_{\mathbb{R}} \to \mathcal{S}$, $k \in \mathbb{N}$, flat base sub-manifold $\mathcal{B}$, size $m \in \mathbb{N}$
**Result:** A generated dataset $\{z_j^*\}_{j=1}^m$ of size $m$

---

**for** $i = 1, \ldots, n$ **do**                                    // Encoding
   $z_i' \leftarrow \varphi(z_i)$
   $w_i \leftarrow$ Proj$(z_i', \mathcal{B})$                     //  $w = $ Enc$(z) = $ Proj$_{\mathcal{B}} \circ \varphi(z)$
**for** $j = 1, \ldots, m$ **do**                                    // Augmenting
   $w_i^* \leftarrow$ Augment$(\{w_i\}_{i=1}^n, \mathcal{B})$
**for** $j = 1, \ldots, m$ **do**                                    // Decoding
   $z_j'^* \leftarrow$ Back-Proj$(w_j^*, \{z_i'\}_{i=1}^n, \{w_i\}_{i=1}^n, k)$        // Algorithm 1
   $z_j^* \leftarrow \varphi^{-1}(z_j'^*)$                     //  $z^* = $ Dec$(w^*) = \varphi^{-1} \circ $ Proj$_{\mathcal{B}}^{-1}(w^*)$
**return** $\{z_j^*\}_{j=1}^m$

---

## C  OMITTED DETAILS FROM SECTION 5

In Section C.1, we provide the details of experimental setup omitted in Section 5, and Sections C.3 to C.5 consists of additional experiments.

### C.1  DETAILS OF EXPERIMENTAL SETUP

In this section, we provide the details of each dataset and other experimental setups with relevant explanations.

**Datasets.**  First, we summarize the details of each dataset in Table 2 and relevant parameters for applying the log-linear model on posets and Algorithm 2.

Table 2: Summary of each dataset and the corresponding setups of Algorithm 2.

| | **Dataset** | | | | | |
| --- | --- | --- | --- | --- | --- | --- |
| | MNIST | CIFAR-10 | Speech Commands | Connectionist Bench | Taiwanese Bankruptcy | Wine Quality |
| **Train Size** | 60,000 | 60,000 | 84,848 | 166 | 5,455 | 5,197 |
| **Test Size** | 10,000 | 10,000 | 4,890 | 42 | 1,364 | 1,300 |
| **Augment Size** | 10,000 | 10,000 | 7,000 | 32 | 1,090 | 1,036 |
| **Class** | 10 | 10 | 35 | 2 | 2 | 7 |
| **Feature** | 784 | 3,072 | 16,000 $\searrow$ 4,000 | 60 | 95 | 11 |
| **Poset** $\Omega$ | $\mathbb{R}_{\geq 0}^{7^2 \times 2^4}$ | $\mathbb{R}_{\geq 0}^{2^{10} \times 3}$ | $\mathbb{R}_{\geq 0}^{2^5 \times 5^3}$ | $\mathbb{R}_{\geq 0}^{2^2 \times 3 \times 5}$ | $\mathbb{R}_{\geq 0}^{5 \times 19}$ | $\mathbb{R}_{\geq 0}^{2^2 \times 3}$ |
| **Base** $\mathcal{B}$ (dim) | $\mathcal{M}_1$ (17) | $\mathcal{M}_5$ (1,410) | $\mathcal{M}_2$ (136) | $\widetilde{\mathcal{M}}_1$ (9) | $\mathcal{M}_1$ (23) | $\mathcal{M}_2$ (10) |
| **Local Data** $\mathcal{D}$ (dim) | $\mathcal{M}_1^*$ (767) | $\mathcal{M}_4^*$ (2,334) | $\mathcal{M}_3^*$ (3,430) | $\mathcal{M}_2^*$ (30) | $\mathcal{M}_1^*$ (72) | $\mathcal{M}_1^*$ (7) |
| **Bandwidth** | 0.05 | 0.05 | 0.05 | 0.05 | 0.05 | 0.05 |
| **Neighbor** $k$ | 8 | 3 | 3 | 2 | 5 | 10 |

Note that MNIST holds a CC BY-SA 3.0 license, CIFAR-10 is released with a MIT license, and Speech Commands is released with a CC BY 4.0 license. Finally, all the UCI datasets (Connectionist Bench, Taiwanese Bankruptcy, and Wine Quality) are licensed under CC-BY 4.0. We now break each group down and explain it in detail:

1. The first group consists of basic dataset statistics. The first three datasets (MNIST, CIFAR-10, and Speech Commands) come with a default train/test split; for the last three UCI datasets, since there is no default train-test split, we take $80\%$ of the whole dataset as the training set, and the remaining $20\%$ as the test dataset. **Augment Size** reports the size of the augmented data, which is roughly $20\%$ of **Train Size**, off by some rounding errors since we assume we augment the same amount of data for each class.

   In all cases, the full training set is used to train the classifier when evaluating the classification performance in Section 5.3, and also to train our data augmentation baseline (i.e., autoencoder) for comparison. The only exception is that for MNIST and Speech Commands, we choose an equal number of samples for every class when doing our pseudo-non-linear data augmentation for implementation convenience.

Finally, due to the extremely high dimensionality of Speech Commands (16000), we down-sampled each data to $4000$ dimensions in the entire experiment due to the computational constraint: solving 16000-dimension convex programs is infeasible in terms of the memory requirement.

2. The second group consists of the poset structure we impose on each dataset when applying the log-linear model for positive tensors. Since our ultimate goal is to utilize the many-body approximation (Eqs. (1) and (2)), by reshaping the feature vector into a high-order tensor, a finer hierarchy of projection can be obtained. Hence, in all experiments, we reshape the feature w.r.t. the prime-number factorization of the number of features. For instance, an MNIST image is in $\mathbb{R}_{\geq 0}^{28 \times 28}$, and we reshape it into a tensor of shape $(7, 2, 2, 7, 2, 2)$, giving it a $6^{\text{th}}$-order tensor structure $\mathbb{R}_{\geq 0}^{7 \times 2 \times 2 \times 7 \times 2 \times 2}$. For notation convenience, in **Poset** $\Omega$, we overload $\mathbb{R}_{\geq 0}^{D}$ to indicate the natural poset structure introduced in Theorem 4.1, and compress the repeated prime factors in the exponent. We note that for the Wine Quality dataset, since the feature dimension is originally $11$, which is a prime, we artificially add $1$ dimension by padding $0$'s, so we get a non-trivial prime factorization.

Next, **Base** $\mathcal{B}$ and **Local Data** $\mathcal{D}$ report the corresponding construction of the base sub-manifold and the local data sub-manifold using either Eq.(1) or Eq.(2) w.r.t. the given poset structure of that dataset. The number in the parentheses reports the corresponding dimension of the constructed sub-manifolds.

3. Finally, **Bandwidth** reports the bandwidth parameter we used when fitting the kernel density model $M$ in the generating step, and **Neighbor** $k$ reports the number of nearest neighbors used in Algorithm 1.

**Classification Performance.** In Section 5.3, we evaluate the augmentation methods by training classifiers and evaluating their prediction accuracy. For MNIST and three UCI datasets, we consider the simple 2-layer MLP, while we use the ResNet-18 for the other two datasets. All models are trained by Stochastic Gradient Descent (SGD) (Ruder, 2016) with learning rate $0.1$, momentum $0.9$, and weight decay $5 \times 10^{-4}$. A step size scheduler is utilized to reduce the learning rate by a factor of $0.1$ every 30 epochs until convergence.

For the two baseline data augmentation methods:

- **Standard method (STD)**: The standard image augmentation methods include random horizontal flipping and cropping. For other modalities, the standard augmentation method corresponds to adding Gaussian noise, which is sampled from $\mathcal{N}(0, \sigma^2)$, where $\sigma$ is chosen to be $1/4$ of the minimum standard deviations over each feature dimension to make sure that the noise level is reasonable.

- **Autoencoder (AE)**: For UCI datasets, we consider a simple two-layer MLP encoder-decoder architecture, where the dimension of the hidden layer is the same as $\dim(\mathcal{B})$ indicated in Table 2, with a bottleneck dimension being $3$. For the image datasets, both the encoder and the decoder are based on convolutional layers. The encoder uses a series of convolutional layers with increasing feature map sizes to progressively downsample the input image, while the decoder mirrors this structure with transposed convolutional layers to reconstruct the image from the latent representation.

  In all experiments, the autoencoder is trained with the Adam optimizer with a learning rate $10^{-3}$ under the mean squared error until convergence.

## C.2 IMPACT OF THE SIZE OF AUGMENTATION

In this section, we conduct additional experiments on the impact of the ratio between the augmented dataset size on the performance. Specifically, we consider the two image datasets (MNIST and CIFAR-10) and the audio dataset (Speech Commands), and vary the size of the original size of the augmented dataset, which is $20\%$ of the original dataset size. The results are presented in Tables 3 to 5.

Table 3: Impacts on the sizes of the augmentation dataset for MNIST.

| AG | 25% | 50% | 75% | 100% |
|---|---|---|---|---|
| **None** | 98.14% | 98.08% | 98.10% | 98.12% |
| **STD** | 92.08% | 92.00% | 92.02% | 92.16% |
| **AE** | 97.91% | 97.86% | 97.90% | 97.97% |
| **PNL** | 98.01% | 98.07% | 97.93% | 97.55% |

Table 4: Impacts on the sizes of the augmentation dataset for CIFAR-10.

| AG | 25% | 50% | 75% | 100% |
|---|---|---|---|---|
| **None** | 89.06% | 89.32% | 88.63% | 88.83% |
| **STD** | 90.42% | 92.00% | 90.60% | 90.92% |
| **AE** | 88.56% | 88.76% | 88.42% | 86.49% |
| **PNL** | 88.72% | 88.32% | 88.53% | 88.67% |

Table 5: Impacts on the sizes of the augmentation dataset for Speech Commands.

| AG | 25% | 50% | 75% | 100% |
|---|---|---|---|---|
| **None** | 83.68% | 83.26% | 84.72% | 84.16% |
| **STD** | 84.34% | 83.84% | 84.98% | 84.25% |
| **AE** | 82.30% | 83.49% | 84.72% | 85.21% |
| **PNL** | 84.20% | 84.61% | 84.56% | 84.43% |

## C.3 SENSITIVITY AND ROBUSTNESS

We examine our proposed method's robustness and sensitivity of the *bandwidth* used when fitting the kernel density model, and also the *number $k$ of the nearest neighbors* used in Algorithm 1. For an easier visual inspection, we use MNIST in this section.

**Bandwidth of Kernel Density Estimation Model.** Consider varying the bandwidth we use when fitting the kernel density model, ranging among $\{0.01, 0.05, 0.1, 0.2, 0.5\}$. The results are shown in Figure 8. We observe that Algorithm 2 is robust under different bandwidths when working with the kernel density estimation model in the generating step.

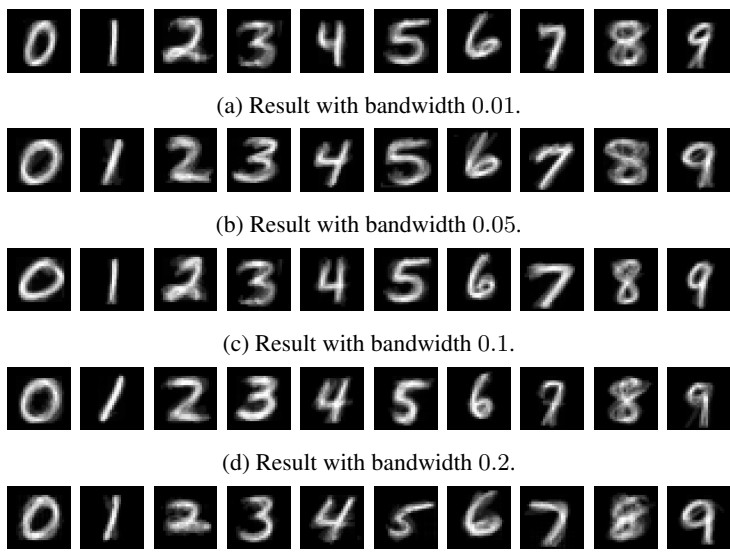

(a) Result with bandwidth 0.01.

(b) Result with bandwidth 0.05.

(c) Result with bandwidth 0.1.

(d) Result with bandwidth 0.2.

(e) Result with bandwidth 0.5.

Figure 8: Augmented data via Algorithm 2 with different kernel density estimation bandwidths.

**Number of Nearest Neighbors.** Next, we consider ranging $k$ among $\{1, 4, 8, 16\}$. The results are shown in Figure 9. Observe that when $k$ is small, e.g., 1, the result of Algorithm 2 tends to overfit since the local sub-manifold $\mathcal{D}$ in Algorithm 1 is defined using only the nearest neighbor. When $k$ goes up, a non-trivial augmentation emerges, robust across different $k$'s.

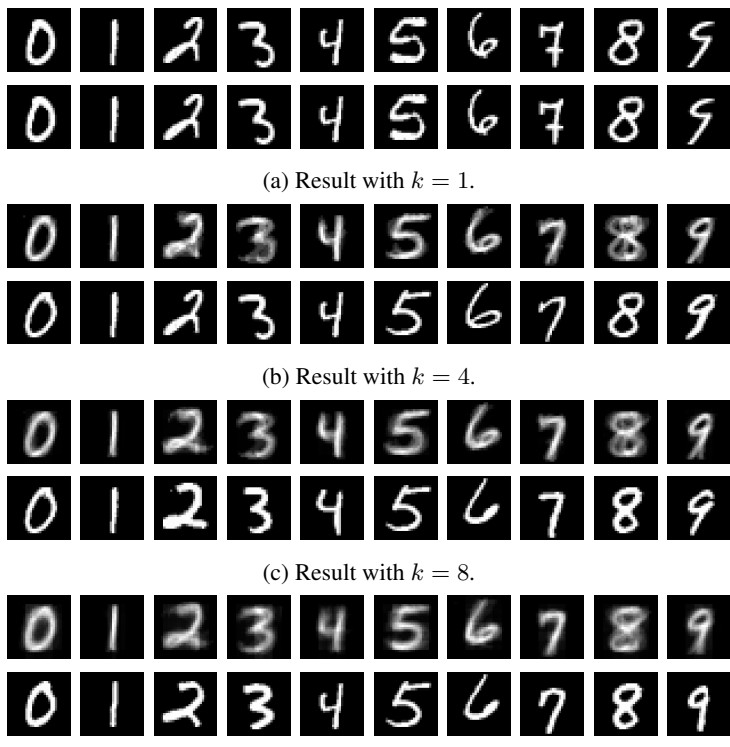

(a) Result with $k = 1$.

(b) Result with $k = 4$.

(c) Result with $k = 8$.

(d) Result with $k = 16$.

Figure 9: (*Top*) Augmented data via Algorithm 2 with different $k$'s for Algorithm 1. (*Bottom*) The closest training data.

## C.4 NECESSITY OF DIMENSION REDUCTION

We demonstrate that dimension reduction, a key building block of our proposed method based on the intuition we have from autoencoder-like models, is necessary for Algorithm 2 to work. For an easier visual inspection, we use MNIST in this section.

**Direct Fitting.** Naive perturbation-based data augmentation methods fall short of high-dimensional data due to the sparsity of the data. Figure 10 shows the results of directly fitting a kernel density estimation model on MNIST with 1000 samples.

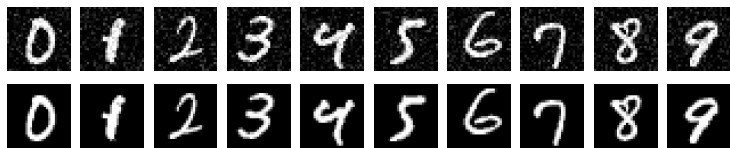

Figure 10: (*Top*) Augmented data via directly fitting a kernel density estimation model with a bandwidth 30. (*Bottom*) The closest training data.

Observe that even with a large bandwidth (30) to introduce variability, we only see a meaningless noisy perturbation on one of the exact training samples, indicating overfitting.

**Local Data Sub-manifold.** A potential problem related to the necessity of dimension reduction is that, if $\mathcal{D}$ captures too much local information about the data (i.e., with low dimension), backward projecting a random latent representation $w^* \in \mathcal{B}$ might already suffice to augment the data in a non-trivial way, without the need for knowing the latent representations of the training dataset. To this end, consider sampling uniformly random latent representations within the empirical range we observed from the latent representations of the training data and perform Algorithm 1. The results are shown in Figure 11.

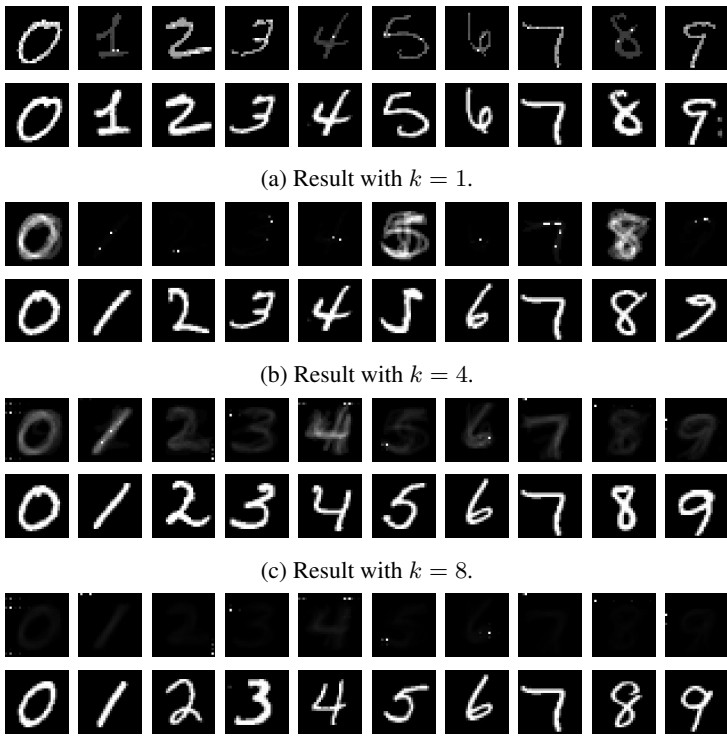

(a) Result with $k = 1$.

(b) Result with $k = 4$.

(c) Result with $k = 8$.

(d) Result with $k = 16$.

Figure 11: (*Top*) Augmented data on random latent representations via Algorithm 2 with different $k$'s for Algorithm 1. (*Bottom*) The closest training data.

For $k = 1$, Figure 11(a) shows that, similar to Figure 9(a), it is possible to overfit one of the training data (i.e., the nearest neighbor of the randomly sampled latent representation). This is not surprising since the base sub-manifold is only of dimension 17, as the random latent representation is sufficiently close to one of the representations of the training data in $\mathcal{B}$, their backward projection result should not deviate too much. Furthermore, we observe the *fading effect*, which intuitively corresponds to *misspecification of the energy*, indicating that the sampled latent representation is fundamentally different from that of the dataset.

As $k$ increases, the benefit of getting informative and meaningful latent representations from the original dataset becomes clear. Specifically, we start to see *degeneration*: from unclear overlappings to collapsing (i.e., only a few pixels are showing). Intuitively speaking, it is because the random latent representation's nearest neighbors appear to be significantly different, hence failing to provide a consistent local data sub-manifold. For instance, in the extreme case when $k = 16$, the local data sub-manifold is completely not informative, resulting in collapsing. Overall, without dimension reduction, we will lose the reference of *realistic latent representations* provided by the original dataset, which leads to bad performance once we are beyond the trivial overfitting regime.

### C.5 CHOICES OF TENSOR STRUCTURE AND CONSTRUCTION OF SUB-MANIFOLDS

In Section 5.2, we consider varying $\ell$ for $\mathcal{B} = \mathcal{M}_\ell$ with the tensor structure $\mathbb{R}^{7 \times 2 \times 2 \times 7 \times 2 \times 2}_{\geq 0}$ on the MNIST dataset. In this section, we further vary the tensor structure as well: in particular, we consider the tensor structure of the MNIST image being $\mathbb{R}^{28 \times 28}_{\geq 0}$, $\mathbb{R}^{7 \times 4 \times 7 \times 4}_{\geq 0}$, and $\mathbb{R}^{7 \times 2 \times 2 \times 7 \times 2 \times 2}_{\geq 0}$.

**Remark C.1.** *For notation convenience, we also write their corresponding poset structures as $\mathbb{R}^{28 \times 28}_{\geq 0}$, $\mathbb{R}^{7 \times 4 \times 7 \times 4}_{\geq 0}$, and $\mathbb{R}^{7 \times 2 \times 2 \times 7 \times 2 \times 2}_{\geq 0}$, and further write the many-body approximation sub-manifold (Eq. (1)) as $\mathcal{M}_\ell(\Omega)$ and its dual (Eq. (2)) as $\mathcal{M}^*_\ell(N, \Omega)$ for a particular poset $\Omega$ to emphasize the dependency.*

Finally, we consider ranging $\ell$ from 1 to at most 4 where we neglect the degenerate case: for instance, in the case of $\mathbb{R}^{28 \times 28}_{\geq 0}$, $\mathcal{M}_2(\mathbb{R}^{28 \times 28}_{\geq 0}) = \mathcal{S}$ as there are only two modes for a matrix, therefore degenerates to direct fitting which is not of interest (see Section C.4). Note that throughout this section, we fix the default local data sub-manifold to be $\mathcal{D} = \mathcal{M}^*_1(N, \mathbb{R}^{7 \times 2 \times 2 \times 7 \times 2 \times 2}_{\geq 0})$ for consistency.

The results for the finest structure, $\mathbb{R}^{7 \times 2 \times 2 \times 7 \times 2 \times 2}_{\geq 0}$, are shown in Figure 12. As $\ell$ grows, the forward projection results (*Top*) preserve the structure of the data better, subsequently improving the quality of the augmented data (*Bottom*). Similar trends can be found in the case of $\mathbb{R}^{7 \times 4 \times 7 \times 4}_{\geq 0}$, as shown in Figure 13.

If we look at the results when using the original matrix structure $\mathbb{R}^{28 \times 28}_{\geq 0}$ (Figure 14), some interesting comparisons can be made. Firstly, if we compare the augmentation results for $\mathcal{B} = \mathcal{M}_1(\mathbb{R}^{28 \times 28}_{\geq 0})$ (Figure 14 (*Bottom*)) with the finer structures counterparts, e.g., (Figure 12(a) (*Bottom*)) for $\mathcal{M}_1(\mathbb{R}^{7 \times 2 \times 2 \times 7 \times 2 \times 2}_{\geq 0})$, one can observe that the results are worse. However, the former requires more dimension ($\dim(\mathcal{M}_1(\mathbb{R}^{28 \times 28}_{\geq 0})) = 55 > 17 = \dim(\mathcal{M}_1(\mathbb{R}^{7 \times 2 \times 2 \times 7 \times 2 \times 2}_{\geq 0}))$) for the base sub-manifold. Similarly, the augmentation results with $\mathcal{B} = \mathcal{M}_1(\mathbb{R}^{7 \times 4 \times 7 \times 4}_{\geq 0})$ (Figure 13(a) (*Bottom*)) also achieve better performance with a lower base sub-manifold dimension.

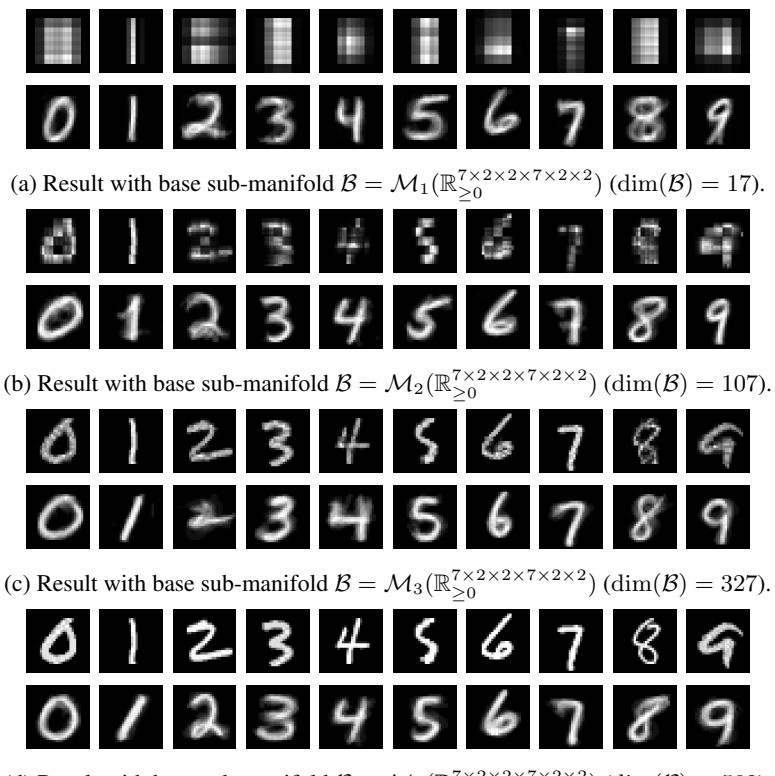

(a) Result with base sub-manifold $\mathcal{B} = \mathcal{M}_1(\mathbb{R}_{\geq 0}^{7 \times 2 \times 2 \times 7 \times 2 \times 2})$ ($\dim(\mathcal{B}) = 17$).

(b) Result with base sub-manifold $\mathcal{B} = \mathcal{M}_2(\mathbb{R}_{\geq 0}^{7 \times 2 \times 2 \times 7 \times 2 \times 2})$ ($\dim(\mathcal{B}) = 107$).

(c) Result with base sub-manifold $\mathcal{B} = \mathcal{M}_3(\mathbb{R}_{\geq 0}^{7 \times 2 \times 2 \times 7 \times 2 \times 2})$ ($\dim(\mathcal{B}) = 327$).

(d) Result with base sub-manifold $\mathcal{B} = \mathcal{M}_4(\mathbb{R}_{\geq 0}^{7 \times 2 \times 2 \times 7 \times 2 \times 2})$ ($\dim(\mathcal{B}) = 592$).

Figure 12: (*Top*) Forward projection on $\mathcal{B} = \mathcal{M}_\ell(\mathbb{R}_{\geq 0}^{7 \times 2 \times 2 \times 7 \times 2 \times 2})$. (*Bottom*) Backward projection on $\mathcal{D} = \mathcal{M}_1^*(\mathbb{R}_{\geq 0}^{7 \times 2 \times 2 \times 7 \times 2 \times 2})$.

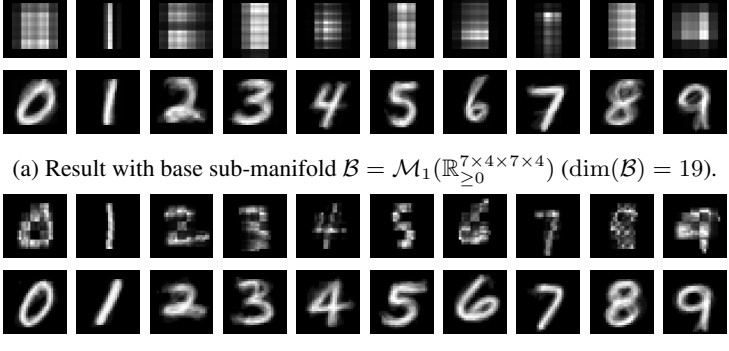

(a) Result with base sub-manifold $\mathcal{B} = \mathcal{M}_1(\mathbb{R}_{\geq 0}^{7 \times 4 \times 7 \times 4})$ ($\dim(\mathcal{B}) = 19$).

(b) Result with base sub-manifold $\mathcal{B} = \mathcal{M}_2(\mathbb{R}_{\geq 0}^{7 \times 4 \times 7 \times 4})$ ($\dim(\mathcal{B}) = 136$).

Figure 13: (*Top*) Forward projection on $\mathcal{B} = \mathcal{M}_\ell(\mathbb{R}_{\geq 0}^{7 \times 4 \times 7 \times 4})$. (*Bottom*) Backward projection on $\mathcal{D} = \mathcal{M}_1^*(\mathbb{R}_{\geq 0}^{7 \times 2 \times 2 \times 7 \times 2 \times 2})$.

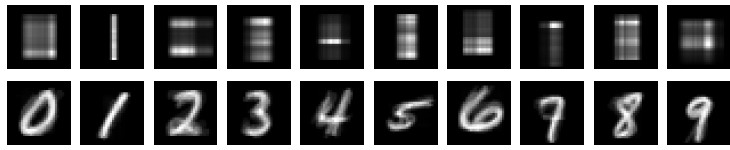

Figure 14: (*Top*) Forward projection on $\mathcal{B} = \mathcal{M}_\ell(\mathbb{R}_{\geq 0}^{28 \times 28})$. (*Bottom*) Backward projection on $\mathcal{D} = \mathcal{M}_1^*(\mathbb{R}_{\geq 0}^{7 \times 2 \times 2 \times 7 \times 2 \times 2})$.

