# OpenReview forum: "Pseudo-Non-Linear Data Augmentation: A Constrained Energy Minimization Viewpoint"
_ICLR.cc/2026/Conference — ICLR 2026 Poster_

### Official Review · Reviewer_gGYH · 2025-10-25

**Soundness:** 3
**Presentation:** 3
**Contribution:** 3
**Rating:** 6
**Confidence:** 3

**Summary:**

This paper proposes a learning-free, geometry-aware framework for data augmentation based on energy-based modeling and information geometry. Unlike deep generative models that rely on trained latent representations, the proposed method constructs a statistical manifold via log-linear models on posets and performs data augmentation through forward and backward projections on low-dimensional submanifolds. The approach, termed Pseudo-Nonlinear Data Augmentation (PNL), combines the linearity of projections with nonlinear geometric curvature, offering both computational efficiency and controllability. Empirical results on images (MNIST, CIFAR-10), speech, and tabular datasets show that PNL performs comparably or better than autoencoder-based and standard augmentation methods, while being faster and interpretable.

**Strengths:**

1. The paper introduces a geometric viewpoint on data augmentation grounded in energy minimization and information geometry, distinct from the dominant deep learning paradigms. The formulation via dually-flat manifolds and log-linear models on posets is both elegant and principled.

2. The method requires no model training, avoiding the heavy computational cost and data dependence of generative augmentation. Projections are formulated as convex programs solvable with first-order methods.

3. Augmentation behavior can be explicitly controlled by the choice of submanifold and the poset structure, enabling fine-grained manipulation of data geometry.

4. The method demonstrates applicability to image, speech, and tabular data without domain-specific modifications.
Shows robustness and consistency, especially on small datasets with high variance.

**Weaknesses:**

1. The baselines are somewhat minimal (autoencoder and standard augmentation).
Comparison to modern generative augmentations (e.g., diffusion-based, mixup, manifold mixup) would strengthen the claims.

2. The method requires defining a partial order (poset) over features, which might be nontrivial or restrictive for unstructured or permutation-invariant data.

3. While the method is theoretically interpretable, more concrete visualization or case studies of controllable augmentation (beyond MNIST/CIFAR) would enhance clarity.

**Questions:**

N/A

---

> ### Author Response · Authors · 2025-11-18
> **Rebuttal**
>
> We thank the reviewer for providing a detailed review and raising actionable suggestions. Below, we provide our point-to-point response.
>
> **(W1)** Minimal baselines comparison.
>
> We have conducted additional experiments on mixup and manifold mixup, where the results can be found in **Table 1**. Together with the new results, our claim does not change: we still consistently achieve competitive, if not the best, performance and robustness upon evaluation.
>
> **(W2)** The method requires defining a poset over features, which might be non-trivial or restrictive.
>
> We would like to clarify first that, in some cases, it is possible to have permutation equivariance within our framework. For instance, for point cloud data represented by the embedding matrix (each row corresponds to a point’s representation/embedding), the canonical lexicographical-index poset structures (i.e., tensor as in **Example 4.1**) with many-body approximation are permutation equivariant **w.r.t. the order of the embedding matrix**. This can be seen from the decomposition. For example, under the two-body approximation of a third-order tensor, a tensor $\mathcal{X}$ is decomposed into three matrices $\mathbf{X}^{(12)}$, $\mathbf{X}^{(13)}$, $\mathbf{X}^{(23)}$ such that $\mathcal{X}\_{ijk} = \mathbf{X}\_{ij}^{(12)} \mathbf{X}\_{ik}^{(13)} \mathbf{X}\_{jk}^{(23)}$.
>
> On the other hand, for the general graph data with (node) features, while it is true that encoding these graphical representations into a poset naturally can not be permutation equivariant since we do not have the rigid matrix/tensor structure to apply systematic many-body approximation, however, we consider this requirement a feature rather than a limitation: when we impose a specific poset structure, some systematic bias is introduced and can be intuitively **controlled**, ensuring the resulting augmentation remains interpretable. Moreover, our method is flexible: if multiple valid posets can represent the data, one can apply the augmentation multiple times with different posets, effectively “diversifying” the encoded bias.
> > A helpful analogy comes from applying PCA to graph data with node embeddings: while PCA on the node embedding matrix can not capture the edge relationship, the resulting latent representation is certainly permutation equivariant.
>
> Finally, regarding the design of posets, our experiments demonstrate that the canonical tensor poset structure can be applied and achieve noticeable performance gain when the data is already structured (e.g., time-series in **Figure 6**, images in **Figure 7**). This provides a good starting point. Suppose practitioners want to incorporate further specific bias/insight/domain expertise related to the particular data modality at hand; our framework can also be easily integrated flexibly, with intuitive guidance on each component.
>
> **(W3)** Additional visualization and case studies of controllable augmentation.
>
> We appreciate the suggestion. We provide extensive visualizations and case studies in the Appendix that illustrate various aspects of our method; due to space constraints, these could not be included in the main text. Moreover, we have added a new, detailed case study in **Section 5.1** to provide more visualization on three case studies, including modalities such as intensity data and time-series data (representing temporally dependent data modalities), and augmentation such as linear interpolation and their energy-model perspective.

---

> ### Author Response · Authors · 2025-11-27
>
> Dear Reviewer gGYH,
>
> Firstly, thank you for your time and thoughtful review. As we approach the end of the rebuttal period, we would like to follow up on the rebuttal of our submission. We appreciate any feedback or clarification you might be willing to share, as your insights are invaluable for helping us address the concerns raised in the review.
>
> If you have the opportunity, we would be grateful for your engagement in the discussion. Please let us know if any additional information from our side would be helpful.

---

> > ### Comment · Reviewer_gGYH · 2025-11-28
> > **Acknowledgment**
> >
> > Thank you for your response. I will keep my score.

---

### Official Review · Reviewer_ZQPF · 2025-10-26

**Soundness:** 2
**Presentation:** 3
**Contribution:** 2
**Rating:** 2
**Confidence:** 3

**Summary:**

This paper proposes a new learning-free data augmentation method that is based on energy-based modeling. The authors first assume a poset structure of the data. Using this poset structure, they map the data to a log linear model on the poset. They then associate dually-flat coordinates with the log linear model. This constitutes the embedding of the data which is then used to generate data augmentations as follows. Given a sub-manifold, the authors project the embedding and then backward project using some heuristic assumptions on the structure of the data. Overall, this results in a task-independent data augmentation method.

**Strengths:**

The paper is presented clearly. The authors manage to put interesting and novel ideas together. This is an interesting addition to the landscape of data augmentation. The fact that the method is learning-free is very appealing.

**Weaknesses:**

1. The poset structure does not fit many data domains well. This is acknowledged by the authors. I don't understand why they suggest that images have an inherent partial order. Directed graphs and time series might conform to such partial orderings.

2. It is unclear what the log linear model captures about the data. The authors mention energy-based geometric modeling but more discussion is needed. It seems that there is first a normalization is done (Example 4.1.) and then this is interpreted as a probability distribution. It is unclear what the meaning of this probability distribution is for many data domains.

3. Apart from the total energy in the features, $\phi(z_i)$ preserves all the information about the data. Similarly, $\theta$ coordinates should then do the same. It is unclear why the geometry here is better for data augmentation than the direct data space.

**Questions:**

1. Can $z_i$ in principle contain data labeling? As far as I understand, $z_i$ is just data without labels. This makes the data augmentation procedure task-agnostic. This results in a method that cannot incorporate any biases that are beneficial for the downstream task except the choice of poset structure. Can you explain how the poset structure introduces such biases in the case of images?

2. Why do you think the probability distribution that comes after applying $\phi$ is a good representation of the data? This should be dependent on the properties of data. For example, could you explain why it is a good representation for images? What is added by the dually flat coordinates? Why would the data be in a linear subspace of this manifold?

3. You cite generative-model-based data augmentation suffers from two challenges: efficiency and controllability. Is this the case for MNIST and CIFAR-10? I believe the challenges apply to harder datasets and I don't see why the proposed method scales to high dimensions.

The following paper is very relevant related work as it has an encoding/decoding scheme over a manifold that is learned.

[1] Yüksel, Oguz Kaan, et al. "Semantic perturbations with normalizing flows for improved generalization." Proceedings of the IEEE/CVF International Conference on Computer Vision. 2021.

---

> ### Author Response · Authors · 2025-11-18
> **Rebuttal (1)**
>
> We thank the reviewer for providing a detailed review with insightful questions, which allowed the authors to clarify our work and improve our presentation subsequently. Below, we provide our point-to-point response.
>
> **(W1, Q2)** The poset structure does not fit many data modalities well.
>
> We thank the reviewer for raising this important point. While it is true that not all data modalities naturally admit a poset structure, we demonstrate that it can nevertheless be applied to, for instance, images, and achieve a non-trivial performance.
>
> Additionally, we would also like to note that it is possible to have permutation equivariance within our framework. For instance, for point cloud data represented by the embedding matrix (each row corresponds to a point’s representation/embedding), the canonical lexicographical-index poset structures (i.e., tensor as in **Example 4.1**) with many-body approximation are permutation equivariant **w.r.t. the order of the embedding matrix**. This can be seen from the decomposition. For example, under the two-body approximation of a third-order tensor, a tensor $\mathcal{X}$ is decomposed into three matrices $\mathbf{X}^{(12)}$, $\mathbf{X}^{(13)}$, $\mathbf{X}^{(23)}$ such that $\mathcal{X}\_{ijk} = \mathbf{X}\_{ij}^{(12)} \mathbf{X}\_{ik}^{(13)} \mathbf{X}\_{jk}^{(23)}$.
> > A helpful analogy comes from applying PCA to graph data with node embeddings: while PCA on the node embedding matrix can not capture the edge relationship, however, the resulting latent representation is certainly permutation equivariant.
>
> On the other hand, for data that does not admit a natural poset structure, however, we consider the requirement of designing a poset a feature rather than a limitation: when we impose a specific poset structure, some systematic bias is introduced and can be intuitively **controlled**, ensuring the resulting augmentation remains interpretable. Moreover, our method is flexible: if multiple valid posets can represent the data, one can apply the augmentation multiple times with different posets, effectively “diversifying” the encoded bias.
>
> Finally, we have also included additional experiments and examples on time series data in **Section 5.1** **Figure 6**, where data comes with a natural poset structure. Moreover, we have incorporated the above discussion into the paper, primarily in **Section 4.1**.
>
> **(W2, Q2)** Confusion on what the log-linear model on poset captures about the data.
>
> In the paper, we deliberately use a specific running example (the positive tensor case) to make the core idea intuitive. However, the natural embedding need not be a normalization; it can be any mapping that defines an appropriate energy function for the data modality under consideration. The probability distribution interpretation (as in the log-linear model) serves primarily as a mathematical formalism to induce this energy function among features, rather than as an object of direct empirical interpretation. What matters in our framework is the energy landscape defined by this model for the data, which governs how data are regularized and augmented.
>
> Previously, due to the space limit, we omitted some parts of the explanation and presented the framework with a very concrete (and visual) example. We have updated the paper to incorporate the above discussion. At the same time, we have added new experiments (**Section 5.1**) to further illustrate the energy landscape (the essence of the log-linear model captures) via the "many-body approximation", which directly controls the complexity of this energy function:
> * 1-body approximation captures energy based *only* on individual feature modes.
> * 2-body approximation model captures *higher-order* feature mode interaction.
> We hope the added discussion and experiments address your concern and potential confusion about the proposed framework.
>
> We have incorporated the above discussion in both **Remark 3.2** and **Section 4.1**.

---

> > ### Comment · Reviewer_ZQPF · 2025-11-27
> >
> > Thank you for your rebuttal.
> >
> > > (W1, Q2)
> >
> > I understand the point regarding controllability but without any poset structure on the data, the method leads to an undesirable systematic bias. I don't see this as a feature but a pure limitation (for the cases where **data does not have the right structure**).
> >
> > I do **not** find the experimental evidences on image datasets non-trivial. The results in Table 1 do not even demonstrate the superiority of well-established baselines with respect to the original dataset.
> >
> > > (W2, Q2)
> >
> > The resulting energy function is over the features and not over data points. I can understand how it makes sense for certain data domains but I don't see the appeal of this viewpoint for, e.g., images. Is this a canonical viewpoint on data such as images?

---

> ### Author Response · Authors · 2025-11-18
> **Rebuttal (2)**
>
> **(W3, Q2)** Confusion on why the geometry induced by our framework is better than ambient space geometry.
>
> We thank the reviewer for this question. The “energy” in our framework serves as a modality-specific potential function that defines the structure of the latent manifold. It can be seen as “MLE of the data” given the energy potential, i.e., the energy models how “stable” or “plausible” a data configuration is under the chosen embedding and poset structure, and our method repeatedly finds the MLE (i.e., the natural/$\theta$-parameters) with the given constraints. This formulation is standard in energy-based models, and can correspond to, e.g., Boltzmann machines.
>
> Another perspective is from the notion of “closeness”: in our framework, the closeness is defined in terms of energy, rather than the naive ambient space distance. We argue that this is a principled and desirable perspective for data augmentation, contrasting with vanilla ambient space feature perturbation augmentation.
>
> We illustrate this with an additional case study in **Figure 5**, where we conduct linear interpolation in both the ambient space and the latent space derived from our framework. We see that, when operating in the energy-aware geometry, we indeed consistently achieve lower energy throughout the interpolation, corresponding to “more probable” data when transitioning from one data set to another. On the other hand, the ambient feature space is ill-conditioned and does not reflect the true intrinsic structure.
>
> Hence, overall, with a well-defined energy for a data modality, our framework offers a principled way to not only obtain meaningful latent representations (and latent space), but also provide a geometrically-aware algorithm for data augmentation. Again, we have incorporated the above discussion in **Section 4.1** for clarity.
>
> **(Q1)** Can our framework incorporate data labeling for task-specific data augmentation?
>
> This is an insightful and subtle question, and we thank the reviewer for raising it. In our framework, $z_i$ indeed represents the latent data embedding and, by default, does not explicitly include label information, hence the procedure is task-agnostic. For classification tasks, we follow the standard in-class data augmentation strategy widely used in the literature, where data points from the same class are augmented together.
>
> Beyond this, our formulation also allows labels to be incorporated directly through the poset structure. Specifically, by treating the class label as the initial element in the poset (which effectively assigns a prior energy to the data), and by designing local data manifolds such that the average of this initial element corresponds to the class label (as in Eq. (2)), we can enforce exact label preservation for the augmented samples.
>
> In the case of images, this design introduces beneficial task-dependent biases: the poset hierarchy can encode relationships among features or semantic components (e.g., spatial parts, channels, or regions) conditioned on the class label, guiding the augmentation to respect class-consistent energy constraints. This controllability allows our method to remain principled yet adaptable to specific downstream tasks. We clarify and incorporate the above discussion in the updated version of the paper.

---

> ### Author Response · Authors · 2025-11-18
> **Rebuttal (3)**
>
> **(Q2)** Why is the probability distribution a good representation of the data?
>
> While the previous responses address this question partially, we would like to provide a standalone answer to unify our thoughts here. In our framework, the specific $\phi$ designed in **Example 4.1** is **not** assumed to be universally suitable for arbitrary data; rather, it should be designed based on the data’s inherent properties, as the reviewer pointed out. More generally, the poset structure and the embedding function $\phi$ should be explicitly constructed to encode prior knowledge about the data modality and to define the form of “energy” one wishes to preserve. This contrasts with purely learning-based data augmentation methods, which rely on implicit priors encoded in model weights.
>
> For instance, if one assumes that images naturally reside in an ambient feature space where simple distance measures (e.g., $\ell_2$) faithfully capture relationships, a straightforward poset with a dummy terminal element connected to each feature dimension suffices. This leads to an energy function conserving total feature magnitudes, and the resulting dually flat manifold essentially coincides with the original feature space.
>
> Hence, to clarify: we do not claim that the provided $\phi$ is universally optimal: it simply illustrates one instantiation of our framework on a very common modality (i.e., tensor-structured data), without any prior knowledge of the actual data modality. In practice, we have experimentally demonstrated that, with this simple canonical example, one can already achieve non-trivial performance gain in both accuracy and stability on downstream tasks with claimed properties (e.g., energy minimization). On the other hand, if a practitioner possesses stronger priors about a data modality, they can design a custom poset and embedding $\phi$ that yields an energy function consistent with those priors; our framework then provides a principled mechanism for energy-consistent data augmentation within that choice. The above discussion is incorporated in **Section 3.2** and **Section 4.1**.
>
> **(Q3)** Why does the proposed method scale to high dimensions?
>
> On the computational complexity side, we discuss this in **Section 4.4**, with additional details in **Appendix A.2**. Our method involves solving a convex optimization problem whose gradients have closed-form expressions and can be computed with cost linear in the data’s *active* feature dimension, i.e., $d_{\text{effective}} = d-d_{\text{constraint}}$, where $d_{\text{constraint}}$ denotes the number of constraints we pose during the information-geometric projection. This allows us to obtain a clean tradeoff between the capacity of the base sub-manifold with the computational complexity. For instance, the 1-body approximation on the $k$-th order square tensors (i.e., each mode has length $\sqrt[k]{d}$) has an active feature dimension of $d_{\text{effective}}$, which scales only at a rate of $O(k \times \sqrt[k]{d})$, making it suitable for computational-restrictive scenarios.
>
> In contrast, diffusion- or flow-based models involve both training and inference stages. Even ignoring training overhead, these models require multiple sampling steps during inference, with each step involving a full forward pass through the learned score/flow networks, resulting in substantially higher computational cost.
>
> With the above discussion, as it is still difficult to exactly quantify the number of iteration required to converge for an projection computation, and also the number of steps required to converge to a good sample during the diffusion inference, we appeal to a direct comparison: as noted in **Footnote 3**, where we point out that most diffusion-based augmentation studies generate only a limited number of samples (≈15) due to the computational cost, which becomes prohibitive for typical large-scale data augmentation tasks like the one considered in our work. In any case, we believe that computation-wise, our method scales to larger dimensions easily with the designed sub-manifolds, as discussed in **Section 4.4**.
>
> Regarding controllability, while recent literature on generative-model-based data augmentation has made progress in conditional generation, these approaches are even more learning-intensive and hence incur significant overhead for training. Our method, by contrast, provides a **learning-free** alternative with light overhead on the controllability, as discussed in **Section 4.4**.
> We will incorporate the above discussion into the updated version of the paper if the reviewer thinks this addresses the concerns.

---

> ### Author Response · Authors · 2025-11-27
>
> Dear Reviewer ZQPF,
>
> Firstly, thank you again for your time and thoughtful review. As we approach the end of the rebuttal period, we would like to kindly follow up on the rebuttal of our submission. We greatly appreciate any feedback or clarification you might be willing to share, as your insights are invaluable for helping us address the concerns raised in the review.
>
> We also noticed that you briefly posted two responses earlier today that later disappeared. We were not sure whether this was intentional or the result of a platform issue, so we wanted to check in case you wished to repost or clarify anything. If you have the opportunity, we would be grateful for your engagement in the discussion. Please let us know if any additional information from our side would be helpful.

---

> ### Comment · Reviewer_ZQPF · 2025-11-27
>
> > (W3, Q2)
>
> As you explained, the energy is over the features. Could you explain what you mean by "MLE of the data," since the potential is not over the data points but over the features? How do you move from an energy function that is specific to a data point to a discussion about the data distribution? I do understand that dually flat coordinates derived from the energy might capture a more nuanced distance compared to Euclidean distance. However, I cannot follow the discussion regarding "stable" or "probable" data configurations, as I assume that requires an energy potential over the data itself.
>
> > (Q1)
>
> Thanks for your answer. It would be an interesting experiment to add label information to the poset structure.
>
> > (Q3)
>
> My question was not about computational complexity but more on curse of dimensionality. I don't see how one can encode the necessary prior knowledge to capture a meaningful manifold of the data, especially as the dimension grows. I understand that the method is learning-free and computationally cheaper.

---

> > ### Author Response · Authors · 2025-11-27
> >
> > > (W3, Q2) *“What does ‘MLE of the data’ mean if the energy is feature-wise? How does this relate to stability or probability?”*
> >
> > Thank you for prompting this clarification, as this helped us refine our terminology. When we referred to “MLE of the data,” we were using the terminology of **information-geometric projection**:
> >
> > - Given a fixed energy potential (poset + chosen interaction order), the forward projection onto an e- or m-flat submanifold computes the point that minimizes the **KL divergence** (dual Bregman divergence) to the embedded sample.
> > - In exponential families, this coincides with a **maximum-entropy / maximum-likelihood projection under linear constraints**, but the constraints are **feature-wise**, not distribution-wise.
> >
> > Thus, when we previously referred to “stable” or “probable” configurations, we meant:
> >
> > - **energy-consistent** within the chosen potential, not
> > - probable under the true (unknown) data-generating distribution.
> >
> > > (Q3) *“How does the method avoid the curse of dimensionality in high dimensions when designing sub-manifolds?”*
> >
> > We agree that this is an important question. In our paper, we offered a practical mechanism for such control is explicitly provided in **Section 4.4**, where the practitioner only needs to choose the poset and the $\ell$-body approximation. These choices determine which interactions are modeled. Moreover, as the dimension grows, the practitioner simply chooses what order of feature mode interactions one wishes to model.

---

> ### Author Response · Authors · 2025-11-27
>
> > (W1, Q2) *“Without a natural poset, the method introduces undesirable bias. Image experiments are not non-trivial.”*
>
> We agree that **if a modality does not naturally admit a meaningful poset**, then any imposed poset introduces a systematic bias. In our framework, this is **intentional rather than incidental**: the poset explicitly determines **which feature interactions are preserved or suppressed** in the latent geometry. Our perspective is that, much like choosing augmentations such as cropping, rotation, or color-jitter, the poset serves as a *declarative way to encode inductive bias*, not as an assumption about the true semantics of the data. More broadly, we note that **all** augmentation methods introduce some form of systematic bias; our method simply encodes this bias via a poset structure, which can be flexible in many modalities.
>
> For images, we fully agree that the positive-tensor poset is not optimal, nor do we present it as a canonical image structure, as we acknowledged in the initial rebuttal. Our claim is only that a *simple, generic poset* can still yield meaningful, interpretable augmentations, as illustrated in **Section 5.2** and **Appendix C**.
>
> Regarding Table 1: our aim in the image experiments was **not** to outperform highly optimized, modality-specific pipelines such as random crop/flip, which encode decades of image-specific engineering. Rather, our goal was to evaluate whether a *generic, learning-free, cross-modal* augmentation can achieve **competitive performance without assuming image-specific transformations**. Indeed, comparing other popular learning-free baselines such as **mixup**, we see that **its performance is even *statistically* worse than ours**.
>
> We appreciate the reviewer’s focus on image results, and we hope the revised draft makes clear that our contribution is *not* to replace classical image augmentation, but to propose a framework that works **uniformly across modalities**, especially in domains where standard augmentations do not exist. As shown across the non-image datasets, our method provides the **most consistent improvement and lowest variance**, which is particularly important in real-world settings where data augmentation is often needed but domain priors are scarce.
>
> > (W2, Q2) *“The energy is over features, not data points. Is this perspective meaningful for images?”*
>
> You are correct that the log-linear model defines an energy over **features**, not directly over **data points**. Our framework does *not* interpret this energy as a canonical or intrinsic distribution of the data. Rather:
>
> - the energy is an **explicitly designed potential** encoding the *inter-feature relationships* the practitioner wishes to preserve or control;
> - the resulting statistical manifold provides a **geometry induced by that potential**, which then guides projection-based augmentation.
>
> For images, the tensor-poset energy reflects **mode-wise structural interactions** (height × width × channel). It is not intended to be a semantic model of image content, but only a **low-level structural prior**, analogous to separable kernels or low-rank tensor decompositions.
>
> We do **not** claim this viewpoint is canonical for images. The framework simply shows that even a simple potential yields coherent augmentations, while richer priors (e.g., hierarchical patches, multiscale structures) can be incorporated by designing alternative posets.
>
> We hope the new **Section 5.1** helps clarify that our method with the positive-tensor poset is *not* proposed as a replacement for classical image augmentations.
>
> ---
> We would genuinely welcome actionable suggestions on how to further clarify the above two questions regarding images in the paper. We suspect some of the miscommunication stems from presentation, rather than disagreement on the technical core. The updated **Section 5.1** (added Nov. 18) provides additional non-image experiments precisely to avoid the impression that we are advancing a canonical structure for images.

---

### Official Review · Reviewer_W952 · 2025-10-31

**Soundness:** 2
**Presentation:** 3
**Contribution:** 2
**Rating:** 4
**Confidence:** 4

**Summary:**

This paper proposes a projection-based data augmentation framework that maps data onto a statistical manifold and performs controlled perturbations through forward and backward projection. The main novelty focuses on achieving learning-free, efficient, and controllable augmentation with semantics preserved, by leveraging the geometric structure induced by posets. Experiments on image and speech datasets validate that the method can maintain core structural features while introducing meaningful variation.

**Strengths:**

The proposed method is learning-free and inverse-consistent. In addition, it is controllable with explicit selection of structural attributes.

**Weaknesses:**

1. Performance gains over simple AE-based augmentation are modest, especially in CIFAR, where improvements are marginal.
2. Efficiency is claimed relative to diffusion models, but no direct comparison or runtime analysis is provided.
3. As discussed in the paper, the method requires data to admit a meaningful poset structure, which cannot directly handle permutation-invariant data such as point clouds or general graphs.
4. The experiments are primarily on images, as argued in line 88, with more different modalities, like graph, video, might be more interesting.

**Questions:**

The backward projection relies on nearest neighbors in the latent space to reconstruct data, which means the augmented samples often stay very close to existing examples. As the dataset grows, will this approach become less scalable and may fail to produce diverse new samples?

---

> ### Author Response · Authors · 2025-11-18
> **Rebuttal (Weakness)**
>
> We thank the reviewer for providing a detailed review and raising actionable suggestions. Below, we provide our point-to-point response.
>
> **(W1)** Performance gains over learning-based baselines are modest
>
> We appreciate the reviewer’s observation regarding the performance gains. We would like to emphasize that one of the key benefits of our proposed method is being *learning-free*, which can be broadly applied across modalities with fine-grained control. Within this scope, our method achieves consistently competitive results compared to the learning-based AE baseline across six diverse datasets spanning image, audio, and tabular domains, covering data scales from thousands to tens of thousands of samples. Furthermore, we have added a new set of experiments covering other popular learning-free baselines, such as mixup, and its learning-based variant, manifold mixup, in **Table 1**. Our method again demonstrates a competitive advantage.
>
> **(W2)** Runtime analysis of the proposed method, besides comparing with diffusion models.
>
> We would like to clarify that the computational complexity of our proposed method is mentioned in **Section 4.4**, with additional details provided in **Appendix A.2**. Specifically, our method involves solving a convex optimization problem whose gradients admit closed-form expressions and can be computed with a cost linear in the data’s *active* feature dimension, i.e., $d-d_{\text{constraint}}$, where $d_{\text{constraint}}$ denotes the number of constraints we pose during the information-geometric projection.
> In contrast, diffusion and flow-based models require iterative sampling procedures during inference, where each step involves a full forward pass through a learned score or flow network, resulting in substantially higher computational cost even when training time is ignored.
> As it is hard to exactly quantify the number of iteration required to converge for an projection computation, and also the number of steps required to converge to a good sample during the diffusion inference, we appeal to a direct comparison: as noted in **Footnote 3**, where we point out that most diffusion-based augmentation studies generate only a limited number of samples (≈15) due to the computational cost, which becomes prohibitive for typical large-scale data augmentation tasks like the one considered in our work.
>
> **(W3)**  Handling permutation-invariant data such as point clouds or general graphs.
>
> We would like to clarify first that, in the case of point clouds, the canonical lexicographical-index poset structures of embedding matrix with many-body approximation are actually permutation-equivariant **w.r.t. the order of the embedding matrix**. This can be seen from the decomposition. For example, under the two-body approximation of a third-order tensor, a tensor $\mathcal{X}$ is decomposed into three matrices $\mathbf{X}^{(12)}$, $\mathbf{X}^{(13)}$, $\mathbf{X}^{(23)}$ such that $\mathcal{X}\_{ijk} = \mathbf{X}\_{ij}^{(12)} \mathbf{X}\_{ik}^{(13)} \mathbf{X}\_{jk}^{(23)}$. On the other hand, for the general graph data with (node) features, while it is true that encoding these graphical representations into a poset naturally can not be permutation equivariant since we do not have the rigid matrix/tensor structure to apply systematic many-body approximation, however, we consider this requirement a feature rather than a limitation: when we impose a specific poset structure, some systematic bias is introduced and can be intuitively **controlled**, ensuring the resulting augmentation remains interpretable. Moreover, our method is flexible: if multiple valid posets can represent the data, one can apply the augmentation multiple times with different posets, effectively “diversifying” the encoded bias. We have updated the paper to incorporate the above discussion in **Remark 4.2**.
> > A helpful analogy comes from applying PCA to graph data with node embeddings: while PCA on the node embedding matrix can not capture the edge relationship, however, the resulting latent representation is certainly permutation equivariant.
>
> **(W4)** Experiments are primarily conducted on images.
>
> We respectfully disagree with the premise that our experiments are "primarily on images." Our evaluation was intentionally designed to demonstrate generality across three distinct modalities:
> - Image (MNIST, CIFAR)
> - Audio (Google-Speech)
> - Tabular, not specific modality (UCI datasets)
>
> However, for illustrative purposes, we indeed appeal to visual modalities, such as images, more than other modalities. To diversify our illustration, we have conducted additional experiments in **Section 5.1** on both synthetic 2D intensive data, as well as time-series data (representing the class of modalities that require temporal dependency).

---

> > ### Author Response · Authors · 2025-11-18
> > **Rebuttal (Question)**
> >
> > **(Q1)** Backward projection often stays very close to existing examples, failing to produce diverse new samples.
> >
> > We would like to clarify two properties of our framework that address the *diversity* problem:
> > 1. *Flexibility and diversity*: In **Figure 5**, we conduct linear interpolation in both the ambient space and the latent space derived from our framework. We see that it is possible to come up with augmentation strategies that produce *diverse* (out-of-distribution even) latent representations, promoting diversity. Such data augmentation strategies can be easily incorporated into our framework in a geometrically aware manner. Moreover, due to the flexibility and intuitive design of both the local data sub-manifold $\mathcal{D}$ and the local base sub-manifold $\mathcal{B}$ (i.e., latent space), we can easily tune the degree of freedom to adjust the need for diversity.
> > 2. *Notion of closeness*: In our framework, the closeness is defined in terms of energy, rather than the naive ambient space closeness. We argue that this is a principled and desirable perspective for data augmentation, contrasting with vanilla ambient space feature perturbation augmentation. More generally, it is possible to produce vastly different samples while it only requires minimal energy from the original sample, which can still be “diverse” in terms of the visual representation: it’s only close in the energy sense, which does not necessarily tie to its ambient space representation.
> > As a concrete visual example, many examples in **Figure 7(e)** have a complicated structure emerging with the simple KDE-based latent sampling already. This, compared to the learning-based baselines such as Auto-Encoder, is arguably much more diverse (looking at the provided code’s AE sampled images, they are visually identical to the training data).

---

> > > ### Comment · Reviewer_W952 · 2025-11-21
> > >
> > > I think the rebuttal is reasonable to me and will raise the rating.

---

> > > > ### Author Response · Authors · 2025-11-22
> > > >
> > > > We thank the reviewer for recognizing our rebuttal and raising the rating!

---

### Official Review · Reviewer_dmdm · 2025-11-01

**Soundness:** 3
**Presentation:** 4
**Contribution:** 3
**Rating:** 8
**Confidence:** 4

**Summary:**

This paper proposes a learning-free and geometry-aware data augmentation framework termed Pseudo-Non-Linear Data Augmentation (PNL-DA). The key idea is to view data augmentation as an energy minimization problem on a latent manifold derived from the intrinsic structure of the dataset.

Instead of using neural generators or search-based augmentation (e.g., AutoAugment), the authors construct an information-geometric latent space and perform constrained energy descent to generate perturbed samples that respect the data manifold while introducing non-linear diversity. This results in augmented data that remain semantically valid yet expand the effective data support.

The paper claims that PNL-DA:
1. requires no additional network training,
2. offers fine-grained controllability over augmentation strength via explicit constraints, and
3. achieves competitive or superior downstream performance across standard image classification benchmarks.

**Strengths:**

1. The paper reframes data augmentation as an energy minimization process under latent-space constraints. This is a refreshing deviation from purely heuristic or search-based augmentations. The geometric viewpoint is conceptually elegant and could inspire follow-up work linking augmentation and manifold regularization.

2. In contrast to most augmentation strategies that involve training auxiliary networks or policy search, this method is deterministic and learning-free once the latent geometry is built. This makes it lightweight and potentially appealing for resource-constrained environments.

3. The energy constraint allows explicit control over perturbation magnitude, which is a desirable property compared to black-box stochastic augmentation policies. The paper demonstrates that tuning this constraint can balance sample diversity and fidelity.

4. Experiments on several vision datasets (CIFAR-10/100, Tiny-ImageNet, SVHN, and others) show that PNL-DA matches or slightly surpasses traditional augmentations and performs competitively against AutoAugment and RandAugment, while using less computational cost.

5. The exposition of the method—particularly the geometric intuition and constrained optimization formulation—is well organized and mathematically sound. The visualizations of latent-space perturbations help the reader grasp the intuition.

**Weaknesses:**

1. The energy constraint is intuitive, but there is no formal proof that minimizing it preserves label semantics or improves generalization bounds. The paper could benefit from a more rigorous analysis linking the energy formulation to risk minimization.

2. The method introduces a constraint coefficient controlling perturbation scale. The results indicate non-trivial sensitivity, but the paper lacks systematic tuning guidelines or ablation to quantify robustness.

**Questions:**

1.The paper frames data augmentation as energy minimization on a latent manifold. Could you clarify the exact connection between your “energy” function and the traditional risk minimization objective? Is the energy equivalent to a potential function, or is it empirically constructed?

2. You define “pseudo-non-linear” transformations. In what precise sense are they pseudo non-linear? Are these transformations non-linear in the data space but linear in latent space, or vice versa?

---

> ### Author Response · Authors · 2025-11-18
> **Rebuttal**
>
> We thank the reviewer for recognizing our contribution and for providing a detailed review. Below, we provide our point-to-point response.
>
> **(W1, Q1)** What is the relationship between energy function, energy minimization, and traditional risk minimization objective?
>
> The “energy” in our framework serves as a modality-specific potential function that defines the structure of the latent manifold. Perhaps a more ML-centric explanation is that it can be seen as “MLE of the data” given the energy potential, i.e., the energy models how “stable” or “plausible” a data configuration is under the chosen embedding and poset structure, and our method repeatedly finds the MLE with the given constraints. This is different from the traditional empirical risk minimization objective for usual model training, which is “MLE of the model parameters” given a finite dataset over some pre-specified data distribution. Hence, we view these two things as fundamentally orthogonal and complementary to each other. We have updated the paper to include further details on this, see, e.g., **Remark 3.2**, **Section 4.1**.
>
> Next, we elaborate on the construction of the energy potential, providing additional examples in addition to the positive tensor examples in **Example 4.1**. Consider time-series data, which is illustrated in the new experiment (**Figure 6**):
> - *Energy Definition*: We can define the energy of a time-series as an intuitive, physical property, such as the sum of its squared amplitudes (proportional to signal power).
> - *Manifold Construction*: We then design an embedding $\phi$ (e.g., $\phi(z_i) = z_i^2$) and a poset (temporal ordering) that encodes this definition.
> - *Augmentation as Minimization*: Our data augmentation then performs *constrained energy minimization*. The forward projection finds a latent representation on a base sub-manifold $\mathcal{B}$ (e.g., a manifold of "low-energy" signals) that is closest in energy (also illustrated in **Figure 5**). The backward projection reconstructs a new sample on the data manifold $\mathcal{D}$ (e.g., preserving the signal's core shape/frequency) that is closest in energy to that latent point.
>
> **(W2)** Lack of systematic tuning guidelines or an ablation to quantify robustness and sensitivity of the proposed method.
>
> We appreciate the reviewer’s observation regarding the constraint coefficient. To address this, we provide in **Appendix C** a set of ablation experiments that systematically evaluate the impact of the coefficient on performance, demonstrating the overall robustness of our method.
>
> **(Q2)** What does “pseudo-non-linear” mean exactly?
>
> The key idea behind our notion of pseudo-non-linearity lies in the interplay between linearity and non-linearity in our projection mechanism. Specifically, the projection is linear when expressed in the intrinsic coordinate system of the statistical manifold $\mathcal{S}$: that is, in the log-domain corresponding to a log-linear model. However, because $\mathcal{S}$ is a curved manifold relative to the ambient data space, these same projections become non-linear when viewed in the original (data) coordinates, as well as in the actual optimization procedure when constraints in the log-domain are presented. This duality, i.e., linear in the manifold’s intrinsic coordinates yet non-linear in the ambient space, motivates our term *pseudo-non-linear* data augmentation. We have incorporated this clarification in **Line 50~52**.

---

> ### Author Response · Authors · 2025-11-27
>
> Dear Reviewer dmdm,
>
> Firstly, thank you for your time and thoughtful review. As we approach the end of the rebuttal period, we would like to follow up on the rebuttal of our submission. We appreciate any feedback or clarification you might be willing to share, as your insights are invaluable for helping us address the concerns raised in the review.
>
> If you have the opportunity, we would be grateful for your engagement in the discussion. Please let us know if any additional information from our side would be helpful.

---

> > ### Comment · Area_Chair_pz4k · 2025-11-28
> >
> > Dear Reviewer,
> >
> > Please make sure you read the authors' response and engage with them in the discussion before the end of the discussion period on **Dec 03 '25 09:00 PM UTC**. This is a hard deadline.
> >
> > Thank you for supporting quality peer review at ICLR.
> >
> > AC

---

### Author Response · Authors · 2025-11-29

Dear AC,

We are aware of the current situation and understand that, at this stage, no further discussion between authors and reviewers will take place. We appreciate the efforts of all reviewers and respect their feedback and suggestions. For this reason, we will conclude our rebuttal with the remarks below, as we feel it would be unfair *to the reviewers* to provide additional arguments when they cannot respond.

## Strengths
1. Reviewers dmdm and gGYH recognize that our method is conceptually elegant, principled, novel, and dmdm mentioned that it has the potential to inspire follow-up research additionally.
2. Reviewers dmdm and ZQPF find the presentation clear, well-organized, visually intuitive, and mathematically sound.
3. All reviewers acknowledge that our method possesses several desirable properties, including being *learning-free*, *efficient*, and *controllable*.
4. Reviewers dmdm and gGYH note that our method empirically demonstrates multiple benefits, including applicability across general data modalities, as well as robustness and consistency in difficult scenarios.

## Weaknesses and Questions
> In line with the above, we do not extend the discussion further here. This section only provides a concise summary of the points addressed during the rebuttal.

1. Model, energy, and applicability
    - Reviewers W952, ZQPF, and gGYH raised concerns about structural limitations. We acknowledged this bias previously in the paper and clarified further in the rebuttal: while the model indeed introduces certain geometric biases, these arise naturally from its learning-free and geometrically intuitive design. At the same time, the framework remains sufficiently general.
    - Reviewers dmdm and ZQPF questioned the notion of energy. We clarified that the energy is defined over features, with dependencies specified by the designed poset structure.
    - Reviewer ZQPF further questioned the applicability and universality of Example 4.1, specifically for image data. We clarified that the example is neither claimed to be universal nor canonical; it simply serves as an intuitive instantiation for illustration. This is potentially a misinterpretation of the claimed contribution and the scope of the framework, given the repeated questions about its suitability for images. As clarified in the rebuttal, the provided examples are illustrative, and no canonical model exists for images yet for this new framework.

    The revised manuscript incorporates these clarifications from the rebuttal, resolves potential sources of confusion, and improves the overall presentation.

2. Additional experiments
    - Reviewer W952 suggested experiments on additional data modalities. We added these results in Section 5.1.
    - Reviewer gGYH recommended evaluating representative baselines such as Mixup and Manifold Mixup. We added corresponding experiments in Section 5.3.
    - Reviewer gGYH also suggested additional visualization and case studies on controllable augmentation. We included these experiments in Section 5.1.

## Remarks

Before the revert, the overall scores ranged from 2–4–6–8 to 2–6–6–8. Specifically:
1. Reviewer W952 increased their score from 4 to 6 after the rebuttal (now reverted).
2. Reviewer gGYH maintained a positive assessment after the rebuttal.
3. Reviewer ZQPF maintained a negative score after the rebuttal (and did not have the opportunity to respond to our latest clarifications).

Finally, we thank all the reviewers for providing the services and engaging in the discussion, as well as the (new) AC, for the extra efforts due to the situation.

Sincerely,
Authors of Submission 16154

---

### Meta-Review · Area_Chair_51hN · 2026-01-05

**Summary:**

Three expert reviewers are quite positive about this paper. However, it seems several key concerns from the Reviewer ZQPF remain not fully addressed. In particular, it does not convincingly justify why imposing an arbitrary poset in domains without a natural partial order (e.g., images) should be considered as beneficial rather than a limitation. The necessity of defining an energy over features rather than over data points remain unclear for image data. Finally, the response does not fully resolve the curse-of-dimensionality concern. However, considering positive feedback from three reviewers, the AC recommend acceptance.

**Reviewer Concerns:**

It seems the comments from all reviewers (except Reviewer ZQPF) have been addressed by the rebuttal. However, the AC belives several key concerns from the Reviewer ZQPF remain not fully addressed. as mentioned in the summary section, the rebuttal does not convincingly justify why imposing an arbitrary poset in domains without a natural partial order (e.g., images) should be viewed as beneficial rather than a fundamental limitation. The necessity of defining an energy over features rather than over data points remain unclear for image data. Finally, the response does not fully resolve the curse-of-dimensionality concern.

**Reviewer Scores:**

I think the three positive reviewers (except Reviewer ZQPF) would give score 6 or above.

---

### Decision · Program_Chairs · 2026-01-26

Accept (Poster)